# A Primer on SO(3) Action Representations in Deep Reinforcement Learning

**Martin Schuck**
Technical University of Munich

**Sherif Samy**
University of Girona

**Angela P. Schoellig**
Technical University of Munich

## Abstract

Many robotic control tasks require policies to act on orientations, yet the geometry of SO(3) makes this nontrivial. Because SO(3) admits no global, smooth, minimal parameterization, common representations such as Euler angles, quaternions, rotation matrices, and Lie algebra coordinates introduce distinct constraints and failure modes. While these trade-offs are well studied for supervised learning, their implications for actions in reinforcement learning remain unclear. We systematically evaluate SO(3) action representations across three standard continuous control algorithms, PPO, SAC, and TD3, under dense and sparse rewards. We compare how representations shape exploration, interact with entropy regularization, and affect training stability through empirical studies and analyze the implications of different projections for obtaining valid rotations from Euclidean network outputs. Across a suite of robotics benchmarks, we quantify the practical impact of these choices and distill simple, implementation-ready guidelines for selecting and using rotation actions. Our results highlight that representation-induced geometry strongly influences exploration and optimization and show that representing actions as tangent vectors in the local frame yields the most reliable results across algorithms. The project webpage and code are available at amacati.github.io/so3_primer.

## 1 Introduction

Accurate reasoning over 3D rotations is a core requirement for machine learning algorithms applied in computer graphics, state estimation and control. In robotics and embodied intelligence, the problem extends to controlling physical orientations through learned actions, e.g., in manipulation policies that command full task-space poses or aerial vehicles that regulate attitude. These tasks rely on trained policies with action spaces including rotations in SO(3).

Dealing with rotations is especially challenging because the underlying manifold is curved, and there exists no minimal parameterization that maps $\mathbb{R}^3$ to SO(3) and is globally smooth, bijective and non-singular. This restriction has led to multiple parameterizations, each with its own trade-offs (Macdonald, 2011; Barfoot, 2017). Euler angles are minimal and intuitive but suffer from order dependence, angle wrapping, and gimbal-lock singularities. Quaternions are smooth and numerically robust with a simple unit-norm constraint, but double-cover SO(3). Rotation matrices are a smooth and unique mapping, but are heavily over-parameterized and require orthonormalization. Viewing SO(3) as a Lie group, one can use tangent spaces, i.e., the Lie algebra m of skew-symmetric matrices, together with the exponential and logarithm maps to represent orientations. Tangent spaces are locally smooth, but globally exhibit singularities at large angles (Solà et al., 2018). Irrespective of the choice of parameterization, any minimal 3-parameter chart must incur singularities, and global parameterizations that avoid singularities are necessarily redundant and constrained.

Applications in deep learning that require reasoning over rotations and orientations have renewed interest in this topic by adding another perspective: irrespective of any mathematical properties, what is the best representation to learn from data in SO(3)? Are low-dimensional representations necessarily more efficient, and does the double-cover of some representations harm training performance? Several works have explored these questions in the supervised setting (Zhou et al., 2019; Peretroukhin et al., 2020; Brégier, 2021). Geist et al. (2024) offer an excellent overview that summarizes the most common representations, links mathematical properties of representations to observed

performance gains, and gives concrete recommendations on supervised tasks like rotation estimation or feature prediction.

What is still missing is a general, systematic evaluation of $SO(3)$ representations in deep reinforcement learning (RL). While the intuitions and results from prior studies on *input/observation representations* for orientations also apply to the RL setting, the most suitable $SO(3)$ *action representation* remains unclear. Action representation requires special attention, as it shapes the exploration dynamics induced by stochastic policies and exploration noise, and has implications for action clipping to comply with actuation constraints. Prior works have proposed specific action representations for a narrow set of problems and algorithms (Alhousani et al., 2023a;b). Schuck et al. (2025a) recently attempted to tackle the general question of best action and observation representations for RL, but limited their investigation to Deep Deterministic Policy Gradients (DDPG) under sparse rewards.

In this paper, we study three widely used continuous-control algorithms: Proximal Policy Optimization (PPO) (Schulman et al., 2017), Soft Actor-Critic (SAC) (Haarnoja et al., 2018), and Twin Delayed Deep Deterministic Policy Gradients (TD3) (Fujimoto et al., 2018). We evaluate action representations under dense and sparse rewards and focus on phenomena specific to RL rather than supervised learning. We show how representations shape exploration, interact with entropy regularization, and affect convergence stability. Finally, we quantify their practical impact across standard robotics benchmarks. In summary, our contributions are as follows:

1. We analyze the most popular RL algorithms for continuous control, **PPO**, **SAC**, and **TD3**, under action spaces that include orientations. These algorithms are extensively used in robotics research to train policies deployed on physical hardware in the real world, and thus are particularly relevant.

2. We investigate why different action representations yield different training performance. Beyond intuitions on properties such as smoothness or uniqueness, we show how observed performance and sample-efficiency differences are attributed to the map between Euclidean network outputs and $SO(3)$. Our analysis highlights the implication of representation-induced action projections on exploration, action scaling, and regularization techniques.

3. We offer concrete guidelines for choosing policy representations and handling representation-induced effects. Building on insights gained in our empirical studies and in three benchmarks on three different robot platforms, we cover algorithm- and representation-dependent pitfalls and how to mitigate them.

This paper aims to make orientation control in RL easy to get right. Consequently, we prioritize clarity, common pitfalls, and ease of implementation over an exhaustive mathematical treatment of manifold optimization. We hope this will help practitioners make a conscious decision on action representations and advance the training of policies with full pose control.

## 2 REPRESENTING $SO(3)$ ACTIONS IN DEEP RL

The set of all 3D rotations forms the Lie group $SO(3)$. It appears in control, graphics, state estimation, and in RL tasks where policies must command orientations, such as manipulation with full end effector pose control or drone control. Multiple representations exist to parameterize $SO(3)$, each with its own properties. In the following, we outline representations, geometric properties, and learning phenomena that matter when projecting neural network outputs to valid rotations and training policies that act on $SO(3)$.

### 2.1 THE $SO(3)$ MANIFOLD

Unlike translation in $\mathbb{R}^3$, which is flat, commutative, and globally parameterized, $SO(3)$ is a compact, curved, and non-commutative manifold $\mathcal{M}$ that admits no global, smooth, minimal chart. Its anisotropic geometry implies bounded, periodic angles and topological constraints that make action representation difficult: minimal coordinates introduce singularities, global coordinates are redundant and constrained, and tangent-space coordinates are only locally valid.

Table 1: Properties of common $\mathrm{SO}(3)$ representations used for actions.

| Representation | $\Delta$ Action | Dim. | Cover | Smooth | Singularities | Constraints |
|---|---|---|---|---|---|---|
| Matrix $\boldsymbol{R}$ | $\Delta \boldsymbol{R}$ | 9 | single | + | - | $\boldsymbol{R}^T \boldsymbol{R} = \boldsymbol{I}$ $\det \boldsymbol{R} = 1$ |
| Quaternion $\boldsymbol{q}$ | $\Delta \boldsymbol{q}$ | 4 | double | + | - | $\|\boldsymbol{q}\|_2 = 1$ |
| Euler angles $(\phi, \theta, \psi)$ | $\Delta(\phi, \theta, \psi)$ | 3 | multi | - | + | – |
| Lie algebra $\mathfrak{m}$ $({}^{\mathcal{E}}\boldsymbol{\tau})$ | ${}^{s}\boldsymbol{\tau}$ | 3 | multi | - | + | – |

Formally defined as

$$\mathrm{SO}(3) = \left\{ \boldsymbol{R} \in \mathbb{R}^{3 \times 3} \mid \boldsymbol{R}^{\top} \boldsymbol{R} = \boldsymbol{I}, \ \det \boldsymbol{R} = 1 \right\}, \tag{1}$$

3D rotations have multiple representations that all map to $\mathrm{SO}(3)$. Table 1 lists a selection of common representations and their respective properties. The Lie algebra $\mathfrak{m}$ denotes the tangent space $T_{\mathcal{E}}\mathcal{M}$ of $\mathrm{SO}(3)$ at the origin. Conversions between tangent increment vectors ${}^{\mathcal{E}}\boldsymbol{\tau} \in \mathbb{R}^3$ and $\mathrm{SO}(3)$ are realized by the capitalized exponential and logarithmic maps $\mathrm{Exp} : \mathbb{R}^3 \to \mathcal{M}$ and its inverse $\mathrm{Log} : \mathcal{M} \to \mathbb{R}^3$. Section A.5 contains example code demonstrating how to realize both $\mathrm{Exp}$ and $\mathrm{Log}$ maps. For an in-depth review on Lie theory and $\mathrm{SO}(3)$ representations, we refer to Solà et al. (2018) and Geist et al. (2024), respectively.

## 2.2 GLOBAL VS DELTA ACTIONS

There are two ways to define orientation actions in deep RL. The most straightforward way of viewing $\mathrm{SO}(3)$ actions is to interpret them as desired orientation in the global frame $\mathcal{E}$. The environment dynamics steer us towards that orientation, e.g., through a low-level controller. In the following, we will call these global actions.

However, the group structure of $\mathrm{SO}(3)$ also permits us to view the action as an intrinsic delta rotation with respect to the current state $s$ of the agent, e.g. for rotation matrices $\boldsymbol{R}_{t+1} = \boldsymbol{R}_t \Delta \boldsymbol{R}_{\Delta a}$ with the delta action $\boldsymbol{R}_{\Delta a}$. Changing the viewpoint makes actions independent from the global frame and thus potentially aids generalization. We will explore the benefits of changing the viewpoint in sections 3 and 4.

## 2.3 MULTI-COVERS, SINGULARITIES, DISCONTINUITIES

Action representations should be unique to avoid multi-modal solutions in policy outputs and targets. Non-injective parameterizations create equivalent actions for the same physical rotation, complicating exploration, entropy regularization, and representation with uni-modal policies. In addition to uniqueness, representations should vary smoothly under small physical rotations. Intuitively, we want representations of actions that lead to similar rotations to lie close in Euclidean space.

Quaternions realize a double-cover from the 3-sphere $\mathcal{S}(3)$ to $\mathrm{SO}(3)$, with $\boldsymbol{q}$ and $-\boldsymbol{q}$ representing the same rotation. Enforcing a hemisphere convention (e.g., nonnegative scalar part) removes the ambiguity but introduces a branch discontinuity on the equator where the scalar part is zero and can cause abrupt sign flips along trajectories.

Lie algebra coordinates use the exponential map $\mathrm{Exp} : \mathbb{R}^3 \to \mathrm{SO}(3)$ which wraps around infinitely often along each axis for $\boldsymbol{\tau} + 2\pi k, k \in \mathbb{N}$. Restricting $\mathfrak{m}$ to a principal branch with angle $\theta = |\boldsymbol{\tau}| \in [0, \pi)$ limits overlap but leaves a cut locus at $\theta = \pi$ where $\log$ is discontinuous and the axis is not unique.

Euler angles are a many-to-one map that is not a fixed $k$-to-1 cover. Most rotations have a unique triple after choosing standard ranges, whereas specific configurations admit infinitely many representations. The classical singularity ("gimbal lock") arises when the first and third rotation axes align and collapse into a combined angle. Independent of this, angle wrapping at $\pm\pi$ introduces discontinuities.

## 2.4 PROJECTIONS

Feedforward policies produce Euclidean outputs that do not satisfy manifold constraints by construction. Rotation matrices must obey $\boldsymbol{R}^{\top} \boldsymbol{R} = \boldsymbol{I}$ and $\det \boldsymbol{R} = 1$, and quaternions must have

unit norm. Therefore, actions must be projected from raw outputs to valid group elements. These projections can be inserted as differentiable layers in the actor, enabling backpropagation almost everywhere. For quaternions, given $\boldsymbol{x} \in \mathbb{R}^4$, normalize $\boldsymbol{q} = \frac{\boldsymbol{x}}{\|\boldsymbol{x}\|}$, which is smooth except at $\boldsymbol{x} = \boldsymbol{0}$. For matrices, the singular value decomposition (SVD) projects $M \in \mathbb{R}^{3 \times 3} = U \Sigma V^\top$ to the closest rotation via

$$\boldsymbol{R} = \boldsymbol{U} \mathrm{diag}\big(1, 1, \det(\boldsymbol{U} \boldsymbol{V}^\top)\big) \boldsymbol{V}^\top, \tag{2}$$

which is differentiable except at degenerate SVDs (Schönemann, 1966). Tangent-space and Euler-angle outputs need no feasibility projection, though magnitudes should be limited to permissible ranges by squashing network outputs through, e.g., $tanh$ activation functions, such that actions are clamped to $|\boldsymbol{\tau}| < \pi - \varepsilon$ and Euler angles to $(-\pi, \pi]$ and $(-\frac{\pi}{2}, \frac{\pi}{2}]$ respectively.

For **TD3**'s deterministic policies, this suffices to guarantee actions lie in $\mathrm{SO}(3)$. Stochastic policies, on the other hand, are often parameterized as multivariate Gaussians. Applying the projection to each sampled action guarantees on-manifold actions, but it also warps the action distribution and renders log probabilities intractable. **PPO** and **SAC** rely on accurate log probabilities, and closed-form corrections for normalization on $\mathcal{S}(3)$ or SVD-based projections are not readily available.

In this paper, we adopt a practical compromise: we project the mean inside the network wherever possible and sample in the ambient Euclidean space. The sampled, off-manifold actions are projected again within the environment to a valid rotation. This approach keeps training compatible with standard log-probability computations while ensuring feasibility at execution time.

### 2.4.1 Unit-rotation Centering

Policy neural networks for RL are designed to initially produce zero-centered output distributions or actions. Counterintuitively, the projections for quaternion and matrix representations map outputs around zero to a wide range of rotation actions (see section A.1.1). This is particularly relevant for delta actions, as the agent must first learn the unit rotation that prevents it from rotating.

One possible remedy is a custom policy network that adds the unit rotation to the action mean. We discuss the performance impact on delta matrix and quaternion representations in section 3.3, and provide ablations on corrections as well as more advanced benchmarks in section A.3. Local tangent vectors and delta Euler angles center around the unit operation and are thus unaffected.

### 2.5 Action Scaling

Orientation control is particularly interesting for physical systems such as robot arms or drones. These systems have bounded angular rates, which motivates policies with limited rotation magnitudes. Rate limits for global orientation targets have to be enforced by an underlying controller or the environment dynamics.

On the other hand, delta rotations in the local frame, as introduced in section 2.2, can scale the increment before mapping to $\mathrm{SO}(3)$. For a tangent vector ${}^{s}\boldsymbol{\tau} \in \mathbb{R}^3$, this is trivially achievable by limiting the output norm. Intuitively, we interpret tangent vectors as vectors attached to the local tangent space of the current orientation. One consequence of viewing $\mathrm{SO}(3)$ actions as delta rotations is that it mitigates discontinuities from wrapping or cut-locus singularities for sufficiently small delta rotations. Delta Euler angles are less straightforward to scale, since the change in orientation magnitude depends on the current orientation. Scaling is either overly conservative or requires a complex chart of orientation-dependent normalizations.

Quaternion and matrix representations can be scaled uniformly via geodesic operations. Using the exponential map presented in section 2.1, $\tilde{\boldsymbol{R}} = \mathrm{Exp}(\alpha \log \boldsymbol{R})$ scales rotations to a maximum angle of $\alpha$, but introduces branch choices and non-smooth points at $\theta = \pi$. In practice, delta actions in the tangent space with norm control provide a well-behaved and straightforward mechanism for action scaling.

## 3 Controlling Pure Orientations

We first study policies that control pure rotations to isolate the effects of action representations. We compare **PPO**, **SAC**, and **TD3** in an idealized environment with only rotational dynamics and

orientation as state. Our analysis tests hypotheses about how representations influence exploration, entropy regularization, and stability, and clarifies what matters for learning SO(3) actions.

## 3.1 ENVIRONMENT SETUP

Formally, we model an episode as a goal-conditioned MDP $\mathcal{M} = (\mathcal{S}, \mathcal{A}, P, r, \gamma)$ with state space $\mathcal{S} = \mathrm{SO}(3) \times \mathrm{SO}(3)$ consisting of states $s_t = (\boldsymbol{R}_t, \boldsymbol{R}_g)$ where $\boldsymbol{R}_t$ is the current orientation and $\boldsymbol{R}_g$ is a goal orientation fixed per episode. Following Geist et al. (2024), we use flattened rotation matrices as observation representations everywhere. Goal-conditioned environments allow us to also analyze sparse reward learning with Hindsight Experience Replay (HER) (Andrychowicz et al., 2017) for off-policy algorithms (**SAC** and **TD3**).

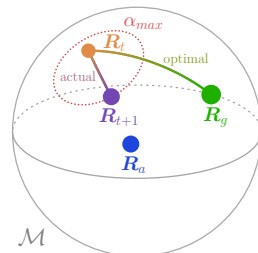

The action space, our object of interest, is configurable. Actions can describe global desired orientations in any of the aforementioned representations or delta rotations. For global actions $\boldsymbol{R}_a$, the deterministic environment transition dynamics are formulated as

$$\boldsymbol{R}_{t+1} = \begin{cases} \boldsymbol{R}_a, & \text{if } d(\boldsymbol{R}_t, \boldsymbol{R}_a) < \alpha_{max} \\ \boldsymbol{R}_t \operatorname{Exp}\left(\frac{\alpha_{max}}{d(\boldsymbol{R}_t, \boldsymbol{R}_a)} \operatorname{Log}\left(\boldsymbol{R}_t^{-1} \boldsymbol{R}_a\right)\right), & \text{otherwise} \end{cases}$$

(3)

with the geodesic distance $d(\boldsymbol{R}_1, \boldsymbol{R}_2) = \arccos\left(\frac{\operatorname{tr}(\boldsymbol{R}_1^\top \boldsymbol{R}_2) - 1}{2}\right)$. Intuitively, equation 3 takes the shortest path towards the desired orientation $\boldsymbol{R}_a$ with a maximum step length of $\alpha_{max}$. Figure 1 visualizes the environment dynamics. Dense rewards $r_t^{\text{dense}} = -d(\boldsymbol{R}_t, \boldsymbol{R}_g)$ are defined as the negative angle to the goal. Sparse rewards are 0 when the angle between state and goal $d(\boldsymbol{R}_t, \boldsymbol{R}_g) \leq 0.1$ and -1 everywhere else. Termination occurs after a fixed step limit of 50.

Figure 1: The agent rotates at max $\alpha_{max}$ radians from the current state $\boldsymbol{R}_t$ to the next state $\boldsymbol{R}_{t+1}$ towards the desired state $\boldsymbol{R}_a$. The goal is to rotate into $\boldsymbol{R}_g$.

## 3.2 PERFORMANCE COMPARISON

We benchmark **PPO**, **SAC**, and **TD3** in the pure-rotation environment using four action parameterizations: rotation matrices, unit quaternions, tangent-space (rotation vectors), and Euler angles, each evaluated as global and delta actions. All other components remain fixed: network architectures, training budgets, observation spaces, and reward definitions are identical across conditions. The results are presented in table 2, with the best results highlighted in blue, and the second-best shown in bold. Results are averaged over 50 runs each. See section A.2 for the training curves, section A.4 for a discussion on Zhou et al. (2019)'s representation, and section A.8 for hyperparameters.

Across algorithms and reward formulations, the delta tangent vector representation almost always results in the best final policy with minor variances between runs. Global matrix representations achieve the second-best performance, except for **SAC** with sparse rewards, where they exhibit poor performance. Other representations often perform poorly, particularly in sparse reward environments, despite using HER.

## 3.3 EXPLAINING THE EFFECTS OF SO(3) ACTION REPRESENTATIONS

The choice of action representations significantly impacts policy performance and training stability. We now dive into the reasons *why* this is the case, and draw conclusions for training policies on SO(3) actions. This section is organized in hypotheses: we state conjectures based on the intuitions from section 2, analyze if these intuitions match our empirical results, and conduct ablations that explain any deviations.

***Hypothesis 1* Smooth, unique representations converge faster and lead to superior policies.**
Our first hypothesis is a common assumption based on the intuition that neural networks better fit smooth functions (Barron, 1993). In addition, the predominant policy network architectures either parametrize a uni-modal distribution (**PPO**, **SAC**) or a single, deterministic action (**TD3**). In this setting, multi-modal action representations should produce conflicting gradients, which harm performance. Consequently, rotation matrices should be the best action representation in SO(3) among the selected ones as they are both unique and smooth.

Table 2: Results for the idealized rotation environment.

|  | **PPO** dense | **SAC** dense | sparse | **TD3** dense | sparse |
|---|---|---|---|---|---|
| $R$ | **-5.4 ± 0.2** | **-4.7 ± 0.3** | -29.4 ± 0.7 | **-4.7 ± 0.2** | **-6.4 ± 0.5** |
| $\Delta R$ | -12.3 ± 1.1 | -5.1 ± 0.3 | -31.0 ± 1.5 | -4.9 ± 0.3 | -20.7 ± 13.4 |
| $q$ | -11.5 ± 1.8 | -5.0 ± 0.5 | -30.2 ± 1.1 | -5.3 ± 0.6 | -9.2 ± 1.5 |
| $\Delta q$ | -22.1 ± 1.8 | -5.0 ± 0.3 | -29.3 ± 0.9 | -5.2 ± 0.4 | -21.6 ± 12.9 |
| $^{\varepsilon}\tau$ | -8.4 ± 0.5 | -7.1 ± 1.5 | -33.5 ± 1.8 | -6.4 ± 0.8 | -30.3 ± 2.2 |
| $^{s}\tau$ | **-5.4 ± 0.2** | **-2.9 ± 0.3** | **-7.9 ± 0.8** | **-3.5 ± 0.3** | **-6.9 ± 0.5** |
| $(\phi, \theta, \psi)$ | -10.8 ± 0.6 | -5.5 ± 0.7 | -35.2 ± 2.1 | -7.3 ± 4.2 | -16.2 ± 3.4 |
| $\Delta(\phi, \theta, \psi)$ | -7.9 ± 0.5 | -5.8 ± 0.5 | **-15.7 ± 8.4** | -7.4 ± 0.9 | -31.2 ± 13.1 |

In our experiments, we see that this is only partially true. The global matrix representation does well in table 2, except for **SAC** and sparse rewards. Based on the smoothness argument, delta matrices should be equally performant but consistently achieve lower performance. The performance difference originates from the fact that delta representations must learn the connection between the current orientation and the goal, instead of only relying on the goal.

While smooth, quaternions display weaker performance due to the double-cover. We can show that for both dense and sparse rewards, the critic learns the multi-modal reward distribution and thus produces conflicting policy gradients if actions are sampled from both hemispheres of $\mathcal{S}(3)$ (see section A.1.6).

The exceptions to our intuition are policies formulated in the tangent space of the local frame. While they are not free of singularities nor discontinuities, the maximum step angle $\alpha_{max}$ limits the policy to a region where the tangent space is unique, has no discontinuities, and the $\mathrm{Exp}$ mapping is almost linear. The singularities and discontinuities at the cut locust are always out of the policy's reach since the space is attached to the local frame. Combined with not requiring projections and a lower dimensionality, local tangent increments outperform global matrix representations even though they must learn the relation between the current frame and the goal.

Tangent vectors in the Lie algebra follow our intuition and produce mixed results, as do global and delta Euler angles due to severe discontinuities and singularities.

> **Conclusion** Uniqueness and smoothness benefit learning, but the properties do not have to hold globally for agents with limited angular step sizes. What matters is that discontinuities and singularities are out of reach for the action space, e.g., as in the local tangent space.

***Hypothesis 2* Representations influence the exploration dynamics on the $\mathrm{SO}(3)$ manifold.**
While the research community has proposed advanced exploration techniques to improve speed of convergence (Houthooft et al., 2016; Plappert et al., 2018), the most common mechanisms are Gaussian stochastic actions (**PPO** and **SAC**) or Gaussian/uniform exploration noise (**TD3**). Samples from these distributions are generally off-manifold. Projecting the perturbed actions back onto their representation manifold (see section 2.4) produces action distributions that can concentrate around small regions of the action space and harm exploration.

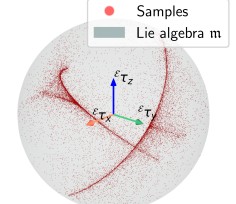

Figure 2: Distribution of Euler angles sampled from $\mathcal{N}(0, 2)$ and squashed with $\tanh$. The resulting orientations are visualized in the 3D Lie algebra $\mathfrak{m}$. Note that samples also lie within the spherical space.

Figure 2 shows this concentration for random Euler angle samples that concentrate around the singularities of the representation. The consequences are most noticeable in the sparse reward environments, where sparse rewards prevent the policy from converging early and agents explore longer (see section A.2). All representations except local tangent vectors and global matrices for **TD3** perform poorly on these tasks.

Additional ablation studies analyzing the replay buffers during training and the distribution of successful goals show that agents' success closely correlates with the exploration distribution. Representations with a more

even spread, i.e., matrix and local tangents, are thus advantageous. See section A.1.1 for a comparison of all distributions, and section A.1.4 for more details on Euler angles.

> **Conclusion** The projection onto $SO(3)$ warps common exploration distributions, which significantly impacts convergence. Euler angles and quaternions are most affected by this, matrices to some degree, and local tangent spaces least.

***Hypothesis 3* Standard entropy regularization leads to suboptimal policies on $SO(3)$.**
As seen above, action projections warp random distributions on the action vector. Apart from exploration, this has consequences for entropy regularization. Maximizing entropy without accounting for the representation manifold may incentivize actions with less randomness after projecting (see e.g. section A.6). Figure 2, where a high-variance Gaussian distribution of angles is mapped to a narrow distribution in $SO(3)$, demonstrates this well. This hypothesis only applies to **PPO** and **SAC**, since **TD3** is missing an entropy term.

The effect is not strong enough for dense rewards to have a strong impact on the results in table 2. However, in sparse rewards, performance for representations with projections on **SAC** is significantly worse compared to **TD3**, whereas the dense variants produce near-identical results. We conclude that the entropy maximization leads to significantly worse exploration behavior than the noise in **TD3**.

To confirm this, we test all action representations with increased entropy coefficients. The full set of experiments for **PPO** and **SAC** can be found in section A.1.7. Our results show that entropy maximization drives actions towards larger norms in Euclidean space. These do not, however, correspond to more random actions because of the maximum step angle $\alpha_{max}$, and instead lead to more stable rotation directions. Scaling delta tangent actions to the range of allowed values mitigates this effect and contributes to the improved performance of **SAC** in section A.1.5. Quaternion and matrix representations do not have a similar mitigation and thus perform poorly.

For Euler angles, elevated entropy levels result in an attraction towards singularities, but entropy coefficients have to be increased by two orders of magnitude to make the effect visible, and thus should not be an issue in standard parameter regimes.

> **Conclusion** Increased entropy regularization drives actions to larger magnitudes, but fails to increase the diversity of matrix and quaternion actions. It influences exploration, particularly in sparse reward environments, but does not change the final policy. Attractions to Euler singularities only become relevant with extremely high entropy bonuses.

***Hypothesis 4* Unit rotation-centered policies improve performance for delta actions.**
Section 2.4.1 remarks that due to the projection onto $SO(3)$, small, zero-centered actions can result in a large spread of rotations for quaternion and matrix representations. Centering the output around the unit rotation should lead to less erratic initial exploration. In addition, agents do not have to learn the group-specific representation of the unit operation, i.e., unit quaternions and rotation matrices.

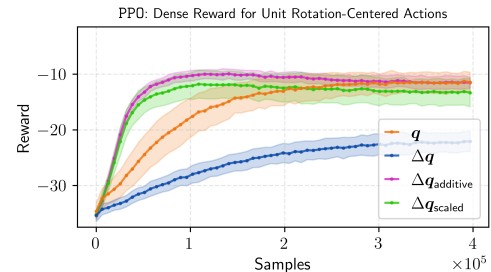

Figure 3: **PPO** learning curves for unit rotation-centered delta actions using delta quaternions.

We test the hypothesis by adding the constant unit rotation to the policy mean. While Figure 3 shows a clear performance improvement over the non-centered delta actions, we find that this mostly holds for **PPO**, and the same modification produces mixed results for **TD3** and **SAC**. For details on the implementation and the complete set of results, including benchmarks from section 4, refer to section A.3.

> **Conclusion** Centering delta actions around the unit rotation improves the performance of quaternion and matrix representations in **PPO**. For **SAC** and **TD3**, the results are not conclusive. Delta tangent and Euler actions are unaffected as they are centered by construction.

*Hypothesis 5* **Scaling actions to the range of permissible rotations boosts performance.** As outlined in section 2.5, tangent vector increments in the local frame can easily be scaled. Restricting the network output to the range of possible rotations should be more efficient because the agent does not need to learn that actions with the same direction and magnitudes larger than $\alpha_{max}$ lead to the same outcome. Furthermore, it removes discontinuities at the cut locus from the action space and allows more fine-grained control in the relevant action ranges.

We can see the effects in all three algorithms. Ablation studies with unscaled tangent increments and dense rewards consistently exhibit a performance difference of around $-1.5$ compared to scaled tangent vectors across **PPO**, **SAC**, and **TD3**. The effect is smallest for **PPO**, where the policy initialization around zero prevents frequent sampling of unscaled actions that reach the cut locus. Agents trained on sparse rewards exhibit a similar decrease in performance. More importantly, however, sparse reward agents sometimes take longer to converge to a near-optimal policy in **TD3**, and in rare cases show degraded performance in **SAC** (see section A.1.5). **SAC**'s failures are caused by the entropy bonus driving actions to regions of the representation with discontinuities and singularities as outlined above.

> **Conclusion** Scaling tangent vectors to the range of permissible angles improves performance and stability. For **PPO**, pay special attention to the log standard initializations.

## 3.4 PRACTICAL RECOMMENDATIONS

We summarize our recommendations for practitioners as follows:

- Prefer delta actions in the tangent space of $SO(3)$. They avoid projection, and for per-step rotations below $\frac{\pi}{2}$ radians, the cut locus and Exp/Log singularities do not affect training.
- Dense rewards can mitigate representation-specific failures while sparse rewards amplify them. Pay special attention to representations in sparse rewards.
- Exploration in the local tangent space is relatively well-behaved. Common strategies like starting with small, zero-centered actions have adverse effects for quaternion and matrix representations.
- Global rotation matrix and quaternion representations can outperform deltas because they need not learn the relation between agent pose and goal. This advantage may vanish when relative object poses matter.
- Delta Euler angles are better than absolute Euler angles, but generally a poor choice.

For additional effects and their explanation, see section A.1.

## 4 ROBOTIC BENCHMARKS

In real-world problems, rotation actions in $SO(3)$ rarely act on purely rotational dynamics as in section 3. Instead, they appear in a larger context, such as controlling a robot arm's pose and gripper to grasp objects, or controlling a drone's orientation and total thrust for trajectory tracking. In this section, we validate the relevance and transferability of our findings by applying the different action parameterizations on three robot benchmarks. Here, we focus on the most promising action combinations, i.e., global matrix and quaternion representations, and delta tangent and Euler angles.

The first benchmark demonstrates the effect of action representations on **PPO** in drone control. We evaluate performance on two tasks: trajectory tracking and drone racing. In the first task, the agent must track a predefined figure-8 trajectory, a standard benchmark in drone control extensively used to compare the performance of different algorithms. For GPU-accelerated training, we use a vectorized version of the safe-control-gym benchmark (Yuan et al., 2022; Brunke et al., 2022). In the second task, we use a similarly adapted version of the IROS 2022 Safe Robot Learning Competition (Teetaert et al., 2025) to train an agent in autonomous drone racing. The aim is to cross four gates as fast as possible in the correct order while avoiding obstacles. **PPO** is the current state-of-the-art for drone control, particularly drone racing, using reinforcement learning (Song et al., 2021; Kaufmann et al., 2023). Following Geist et al. (2024), we convert all $SO(3)$ observations to

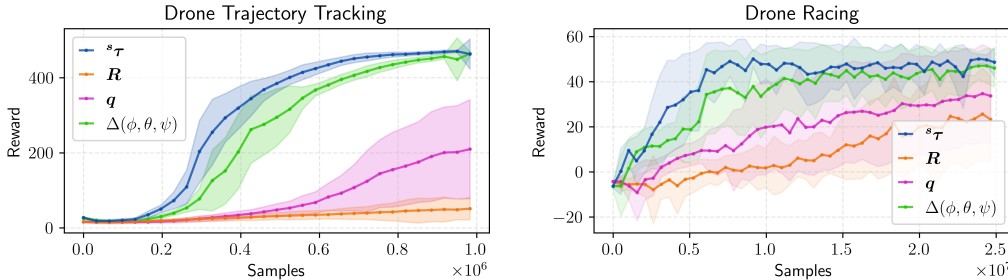

Figure 4: Achieved reward for the trajectory tracking (left) and drone racing competition (right) tasks across action parameterizations. Shaded areas indicate the standard deviation across 25 seeds.

rotation matrices. As in section 3, we only change the action representation used by the policy. For details on the choice of hyperparameters, refer to section A.8.

In figure 4, action representations significantly impact convergence speed on both benchmarks. Actions in the local tangent space consistently outperform other representations by converging faster and achieving higher rewards. Surprisingly, Euler angles are second before rotation matrices and quaternions. The reason is the limited range of angles required in the tasks. The drone cannot deviate too much from the upright orientation without crashing, and hence the policy remains in a region of $SO(3)$ where Euler angles are still well-behaved. In contrast, absolute quaternion and matrix actions are highly random at initialization (see section A.1.1), which leads to fast crashes and limits progress. Ablations in section A.3 show that for these unstable environments, unit-centering (see section 2.4.1 and hypothesis 4 in section 3.3) is a key factor for performance. It is highly advisable to use relative and unit-centered representations this class of systems.

Next, we benchmark action representations on RoboSuite (Zhu et al., 2020), a simulation framework implementing a suite of manipulation environments leveraging MuJoCo (Todorov et al., 2012). The reference baseline uses **SAC** with shaped rewards on nine tasks spanning from single-arm block lifting to complex tasks such as peg-in-hole tasks with two arms. As in section 3, we only change the action representation used by the policy. We follow the requirements outlined in the benchmark, training for 5M steps across five random seeds each using operational space control (Khatib, 2003) to convert from policy actions to joint torques. Figure 5 presents the results. We report the mean performance and standard deviation as a fraction of the maximum achievable reward in percent.

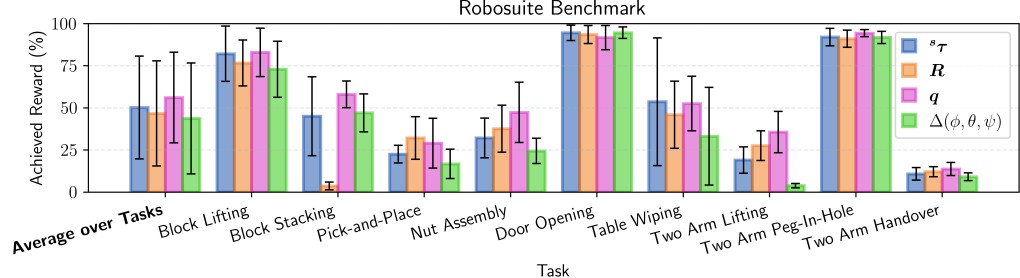

Figure 5: Achieved reward across the RoboSuite benchmark as a fraction of the maximum possible reward. Error bars denote the standard deviation across five seeds.

As expected, global actions do well on **SAC** with dense rewards. The dense reward compensates for exploration issues. Notably, quaternions outperform the matrix representation on several tasks. Contrary to section 3, local tangent actions, while competitive on most tasks, do not exceed the performance of quaternions. Global actions may benefit tasks requiring the arm to move into a few select poses. Overall, the narrow performance gaps between these representations within the same task and larger performance gaps between tasks indicate that other factors, such as reward design and overall task difficulty, dominate the benchmark.

In the last benchmark, we adapt the setup from Andrychowicz et al. (2017) for goal-conditioned robot arm control to include pose goals. We extend the agent's action space to include the gripper orientation and employ HER with **TD3**, the successor of the previously used DDPG (Lillicrap et al.,

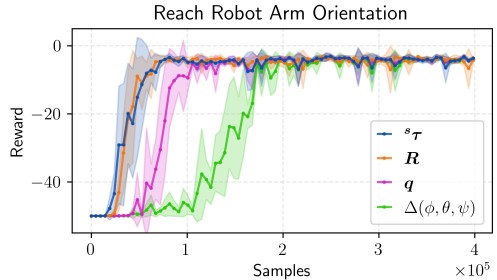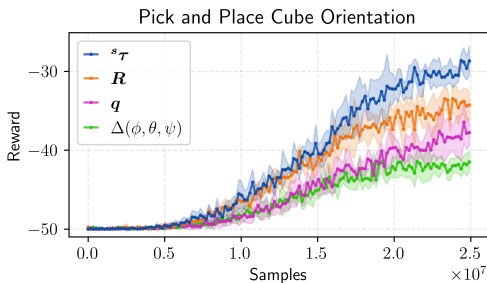

Figure 6: Achieved reward on `ReachOrient` (left) and `PickAndPlaceOrient` (right). Both tangent and matrix action representations converge fast for the reach task, with quaternions second and Euler angles last. On the harder pick and place task, the local tangent space representation significantly outperforms other representations both in performance and convergence speed.

2016). Analogous to the original reach and pick and place tasks, we define a reach task where the agent must lift its end effector into a target position and orientation, and a pick and place task where the goal is to place the cube into a randomly sampled target position and orientation. The remaining fetch tasks, sliding and push, cannot easily be modified to include orientation goals and are thus omitted. For more details on the environment design, refer to section A.7.

As in the previous benchmark, we analyze switching the $SO(3)$ action parameterizations of the policy. The results, averaged over five runs, are shown in figure 6. Policies with matrix and tangent representations quickly converge to a near-perfect policy, quaternions follow slightly delayed, while Euler angles significantly lag. On the second task, the tangent representation again clearly outperforms other representations at $69.8\%$ success rate, with matrix second at $54.1\%$, quaternions third at $46.7\%$, and Euler angles last at $32.3\%$. Here, the combined randomness of the initial cube and target orientations requires the policy to cover a significant part of $SO(3)$. Consequently, the differences between representations become more pronounced, as seen by the 2x increase in success rates between Euler and tangent representations.

## 5 CONCLUSION

Action representations on $SO(3)$ shape exploration dynamics, entropy rewards, and smoothness of policies in deep RL. In this paper, we established that the choice of representations alone impacts performance and convergence speed. We analyzed the behavior of popular representations for **PPO**, **SAC**, and **TD3**, showed how the choice of representation affects learning, and gave clear recommendations for practitioners looking to train agents for orientation control. Delta actions in the tangent space offer the best performance, especially for tasks that cover all or a large subset of $SO(3)$. While there are viable alternatives for specialized domains (e.g., where only a few fixed orientations are required and global quaternion and matrix representations can be competitive, or with orientations close to identity, where Euler angles are an alternative), the tangent space overall serves as an efficient general representation across algorithms and tasks.

One limitation of this paper is its restriction to state-based observations and small networks. Observations do not affect the action space; thus, our results likely still apply. However, this requires empirical evidence. In addition, we did not consider discrete action space algorithms, which open up entirely new questions, e.g. on discretization schemes and the required density of the $SO(3)$ cover. We also noticed a lack of suitable benchmarks that require control over the full $SO(3)$ manifold. Our extension to the HER environments could act as a starting point to build up a standard set of tasks focusing on this capability. Finally, diffusion policies have found widespread adoption for imitation learning in robotics. A similar study on suitable representation choices for diffusion may reach significantly different conclusions due to diffusion models' multi-modality and noise processes.

### 5.1 REPRODUCIBILITY STATEMENT

The code to reproduce our experiments is publicly available at github.com/amacati/so3_primer. It includes the training scripts and instructions for running the experiments we describe.

### 5.1.1 ACKNOWLEDGMENTS

We thank SiQi Zhou and Marcel Rath for their helpful feedback on the manuscript, and Jan Brüdigam for the discussions on SO(3) representations and their impact on RL training. This work was supported by the Robotics Institute Germany under BMBF grant 16ME0997K, and by the Humboldt Professorship for Robotics and Artificial Intelligence.

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

# A APPENDIX

## A.1 ADDITIONAL EFFECTS

This section discusses additional effects and presents extended ablation studies related to section 3.3.

### A.1.1 PROJECTIONS OF NOISE SAMPLES

The properties of each representation affect the mapping of Euclidean noise samples when projected onto $\mathrm{SO}(3)$. We showcase how noise samples from identical distributions result in entirely different rotation distributions when projected. Then, we evaluate the effect of projecting action samples on **PPO** and **SAC**, two algorithms that utilize probability densities computed based on a Gaussian assumption.

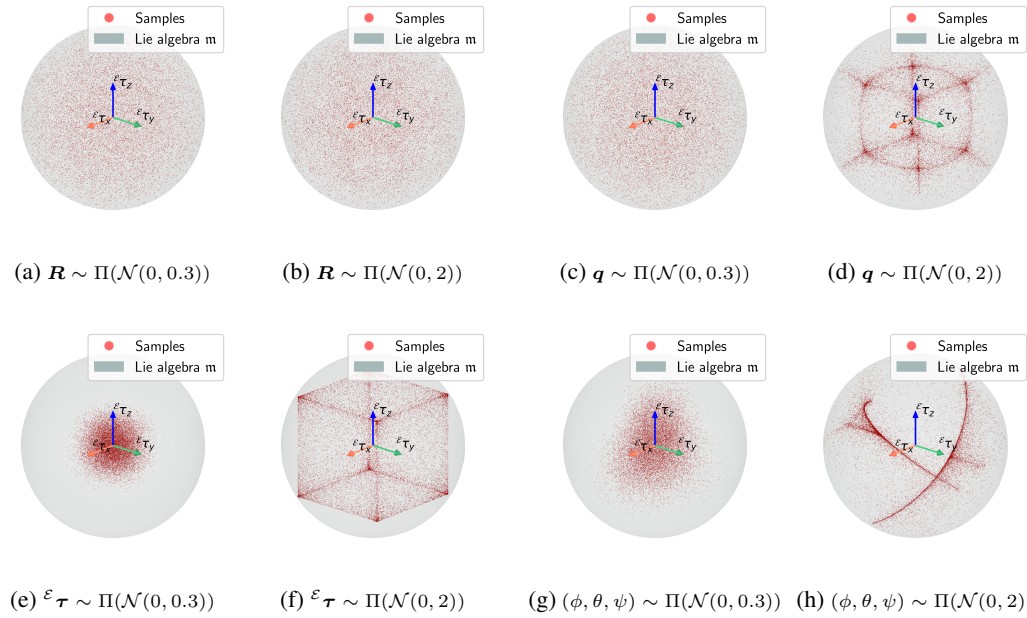

(a) $\boldsymbol{R} \sim \Pi(\mathcal{N}(0, 0.3))$     (b) $\boldsymbol{R} \sim \Pi(\mathcal{N}(0, 2))$     (c) $\boldsymbol{q} \sim \Pi(\mathcal{N}(0, 0.3))$     (d) $\boldsymbol{q} \sim \Pi(\mathcal{N}(0, 2))$

(e) $^{\varepsilon}\boldsymbol{\tau} \sim \Pi(\mathcal{N}(0, 0.3))$    (f) $^{\varepsilon}\boldsymbol{\tau} \sim \Pi(\mathcal{N}(0, 2))$    (g) $(\phi, \theta, \psi) \sim \Pi(\mathcal{N}(0, 0.3))$   (h) $(\phi, \theta, \psi) \sim \Pi(\mathcal{N}(0, 2))$

Figure 7: Samples from a squashed Gaussian distribution projected onto the manifold using the projections $\Pi$ outlined in section 2.4. Each action representation has its own characteristic distribution after sampling. Samples are visualized as 3D points in the sphere of the Lie algebra $\mathfrak{m}$.

Figure 7 depicts the projection of noise sampled from squashed Gaussian distributions onto $\mathrm{SO}(3)$, visualized in 3D through the Lie algebra $\mathfrak{m}$. The rotation matrix projection yields a distribution almost independent of the noise level, resembling a uniform distribution. Tangent vectors remain normally distributed for small noise magnitudes, but concentrate at the boundaries of the 3D tangent space because of saturation. Quaternions display a uniform distribution similar to rotation matrices for small noise levels. Indeed, without clipping, projecting zero-centered Gaussian noise leads to a uniform cover of $\mathrm{SO}(3)$. At larger noise levels, the distribution of rotations concentrates along a narrow sub-manifold. Finally, Euler angles act similarly to tangent vectors at smaller noise magnitudes, but concentrate around the two curves corresponding to the singularities for larger magnitudes.

For stochastic policies used in **PPO** and **SAC**, a natural question is whether projecting sampled actions to the valid representation manifold speeds up convergence, as the critic never sees off-manifold actions. Ablation runs in figure 8 show this is false. Projections lead to a significant performance loss in **PPO**, because the probability ratios $\frac{\pi_\theta(\mathbf{a}|\mathbf{s})}{\pi_{\theta_{old}}(\mathbf{a}|\mathbf{s})}$ used in its clipped surrogate objective no longer matches the ratio of the unprojected action. Newly computed probabilities use projected actions, while the stored probabilities of the old policy are based on non-projected actions. **SAC** remains unaffected as action probabilities are re-computed online during policy updates, with projections applied afterwards. However, we do not observe any performance improvements.

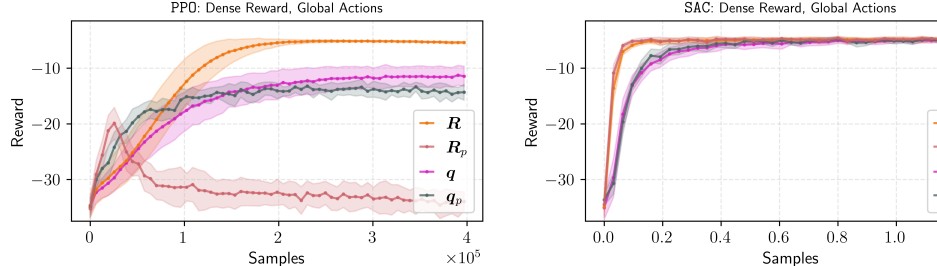

Figure 8: Learning curves for **PPO** (left) and **SAC** (right) evaluating the effect of projecting action samples onto the manifold for the global quaternion and rotation matrix action representations using dense rewards in the idealized environment.

Based on our ablations, we recommend not projecting the actions of **PPO** and **SAC** after sampling, and instead relying on the environment for the action projection. Introducing projections offers no performance gains in the best case, and has the potential to severely harm convergence.

### A.1.2 EULER ANGLE NONLINEARITIES AND DISCONTINUITIES

Euler angles show very inconsistent performance across algorithms and benchmarks. This section analyzes why this is the case and relates their performance to the nonlinearities and discontinuities outlined in section 2.

In the idealized environments, Euler angles perform poorly almost everywhere. The only exception is **PPO** with dense rewards. The reason is that the relation between Euler angle rates, and by extension delta Euler angles, of a fixed magnitude and the induced angular change in orientation becomes increasingly nonlinear, making it harder to learn orientation control at orientations far from the identity.

In **PPO** with dense rewards, the network initialization has to be adjusted to produce incremental actions narrowly focused around zero at the beginning of the training. Otherwise, agents do not converge to a successful policy. Adapted initialization partially alleviates the issue and leads to a suboptimal, but stable policy.

While the delta Euler angle chart is highly nonlinear at large angles, it is almost identical to the incremental tangent representation at small changes around the identity. We can see this in the drone control benchmarks in section 4, where $\Delta(\phi, \theta, \psi)$ performs nearly as well as $^s\tau$. Stable drone flight only requires states in a small region of $\mathrm{SO}(3)$, and the angular changes in the trajectory task are minimal. Hence, the performance of Euler angles is not surprising. In the drone racing task, drones are flying more aggressively, and thus the performance gap to $^s\tau$ increases.

In the RoboSuite benchmark, performance varies significantly, from being worse than other representations to having minor differences. We attribute this to varying levels of pose control required to solve the tasks. On `ReachOrient` and `ReachPickAndPlace`, which cover a large subset of $\mathrm{SO}(3)$, $\Delta(\phi, \theta, \psi)$ is again the worst among all policies.

Practitioners should avoid Euler angles for $\mathrm{SO}(3)$ action representations. Delta Euler angles can be successful on tasks that only require small angular changes, because this avoids the heavy nonlinearities and singularities at larger angles. Examples are drone control at moderate speed and manipulation with minor orientation adjustments. However, they offer no advantages over local tangent increments $^s\tau$, and yield worse performance as the required coverage of $\mathrm{SO}(3)$ increases.

### A.1.3 PROJECTING **SAC**'S ACTOR OUTPUT

**SAC** uses a squashed Gaussian policy parameterization, which prevents the projection of the mean onto the manifold of its representation. Samples are generated using

$$\pi_\theta(\mathbf{s}) = \tanh(\mathbf{u} \sim \mathcal{N}(\boldsymbol{\mu}_\theta(\mathbf{s}), \boldsymbol{\sigma}_\theta(\mathbf{s}))). \tag{4}$$

Importantly, the squashing is applied *after* sampling. Consequently, projecting the mean *before* sampling will reduce the range of mean values the distribution can sample from. E.g., normalized Euler angles of $[0, 0, 1]$ as mean will become approximately $[0, 0, 0.76]$, which makes reaching some orientations in $\mathrm{SO}(3)$ infeasible. Projecting actions *after* sampling does not improve performance as shown in appendix A.1.1. Therefore, we keep **SAC**'s actions completely off-manifold.

Combining off-manifold mean actions and **SAC**'s maximum entropy formulation results in several entropy-related issues discussed in section A.1.7 with quaternions and rotation matrix representations. However, we adhere to the commonly used policy parameterization for **SAC** due to its widespread use, squash actions once after sampling, and rely on the environment to correctly project the policy's actions.

The mean actions can be projected in **PPO** and **TD3** as samples are left unbounded (**PPO**) or use additive exploration noise with clipping (**TD3**).

### A.1.4 Exploration with Euler using Delta Actions

Delta Euler angles outperform their global counterparts, but struggle in tasks that require full orientation control. Here, we show how wrapping Euler angles around the singularities at $\theta = \pm\pi/2$ dominates exploration behavior during training and leads to a poor coverage of $\mathrm{SO}(3)$.

Starting from an orientation whose pitch angle $\theta_0 \approx 0$ and sampling normally distributed actions from $\mathcal{N}(0, \sigma)$, the distribution of the next pitch angle $\theta_1$ remains approximately normally distributed. However, if $\theta_0$ is non-zero (e.g. $\theta_0 = \frac{\pi}{4}$), samples exceeding $\theta = \frac{\pi}{2}$ will wrap around. Thus, the density of $\theta_1$ in $\left[\frac{\pi}{2}, \frac{3\pi}{4}\right]$ adds to that of $\left[\frac{\pi}{4}, \frac{\pi}{2}\right]$. Therefore, the probability density of $\left[\frac{\pi}{4}, \frac{\pi}{2}\right]$ outweighs that of $\left[0, \frac{\pi}{4}\right]$. Hence, the agent is more likely to move closer to the singularity rather than away from it. This effect continues until the singularities, where one half of the Gaussian distribution is mirrored onto the other half as shown in figure 9.

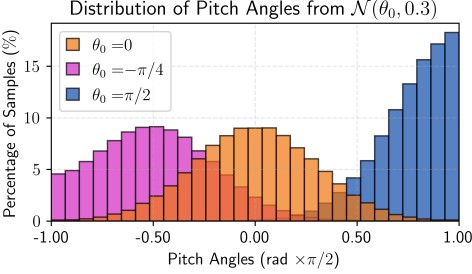 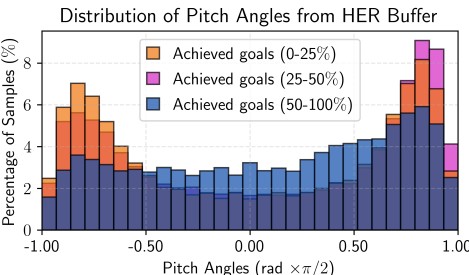

Figure 9: Distribution of pitch angles sampled from Gaussian distributions with different means and a standard deviation of 0.3 (left) and from the achieved goals stored in the HER replay buffer during training using **SAC** with delta Euler angles and sparse rewards in the idealized environment (right).

In the absence of a dense reward that provides immediate feedback to policies about the quality of their actions, this effect severely slows down convergence and results in the high variance displayed by both **SAC** and **TD3** for sparse rewards in section A.2. Since policies are initialized randomly and rely on noise for exploration, a large percentage of their initial trajectories end up near the singularities following this random walk. Accordingly, by re-labeling experiences through HER, they learn to reach these singularities consistently.

However, due to the highly nonlinear and discontinuous behavior of Euler angles near these points, policies fail to explore further regions of the goal space, resulting in the observed results for **SAC** and **TD3**. By inspecting the distribution of pitch angles for achieved goals stored in the replay buffer for **SAC** in figure 9, it is clear that points near the singularities dominate the first and second quarters of the training process. Only later on, during the second half of the training process, does the distribution of goals become more balanced.

### A.1.5 SCALING TANGENT VECTORS

Section 2.5 outlines how tangent vectors can be scaled to only encompass the range of possible rotations with a limited angle $\alpha_{max}$. In section 3.3, we explain how this helps avoid the cut locus of the tangent space at large action norms. Here, we present the training curves of our ablations in figures 10, 11 and 12. As previously stated, the final mean reward of scaled policies is slightly increased by 1.5 to 2 due to more fine-grained control. Much of the unscaled policies' action space lies outside the $\alpha_{max}$ limit and thus maps to the same actions.

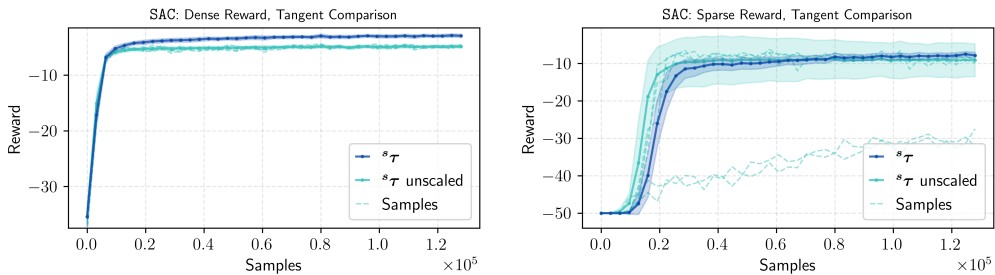

Figure 10: Learning curves of the scaled and unscaled incremental tangent vector representation for **SAC**. We show the worst five among 50 unscaled runs to emphasize that the increased variance stems from a small number of runs that fail to make significant progress.

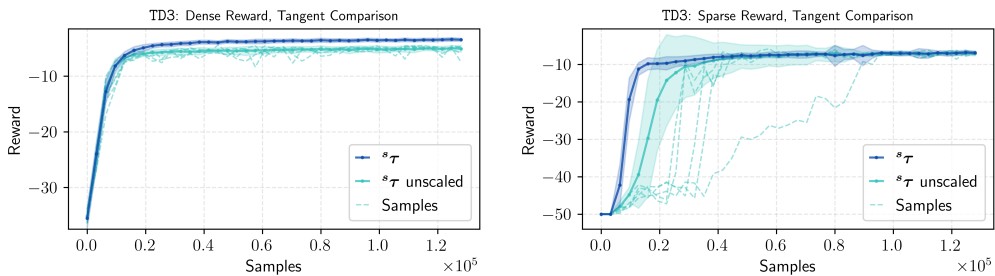

Figure 11: Learning curves of the scaled and unscaled incremental tangent vector representation for **TD3**. Again, we show the worst five among 50 unscaled runs to demonstrate that the increased variance stems from a small number of runs with significantly slower convergence rates.

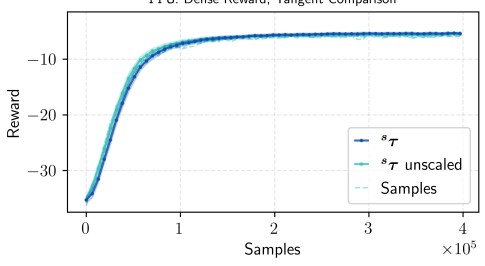

Figure 12: Learning curves of the scaled and unscaled incremental tangent vector representation for **PPO**. As before, we show the worst five among 50 unscaled runs. Dense rewards prevent convergence issues in **PPO**.

The effect is smallest for **PPO**, where the initialization of the policy leads to actions close to zero. For **SAC** and **TD3**, the effect is slightly more visible in dense environments. Agents in sparse environments infrequently converge with significant delay (**TD3**) or fail to converge (**SAC**), causing

the increase in variance across runs. The entropy bonus drives actions towards larger magnitudes around the cut locust before learning a reasonable policy, which causes the complete failures in **SAC**. Dense rewards prevent this by balancing the entropy bonuses with a continuous signal towards successful policies from the beginning.

### A.1.6 CONFLICTING POLICY GRADIENTS

Given the uni-modal policy parameterization used by all three studied algorithms, the full double-cover of quaternions and partial overlap of the Lie algebra $\mathfrak{m}$ for $\|\boldsymbol{\tau}\| > \pi$, harm the learning process. Multiple optimal actions produce conflicting gradients that pull the policy in different directions. In the case of the quaternion double-cover, these actions point in opposite directions, yielding opposing gradient signals during policy updates.

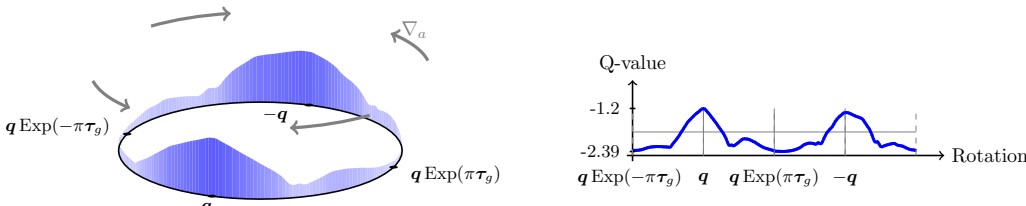

(a) **SAC** Q-values along the double-cover of quaternions for sparse rewards.

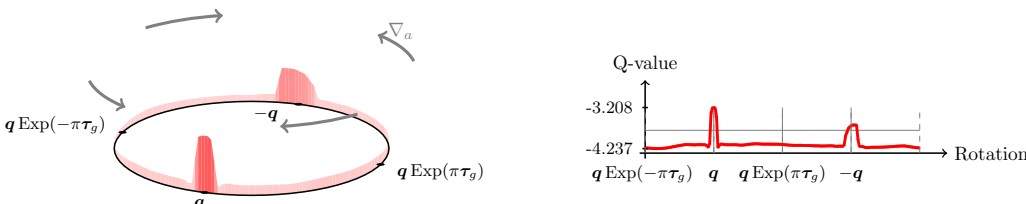

(b) **TD3** Q-values along the double-cover of quaternions for sparse rewards.

Figure 13: Learned Q-values of **SAC**'s and **TD3**'s critics for sparse rewards using the global quaternion action representation in the idealized environment. We sample equally-spaced actions discretely along the geodesic connecting $\mathbf{q} = \pi(\mathbf{s})$ and $-\mathbf{q}$ that passes through the goal orientation on the $S(3)$ manifold by applying the $\mathrm{Exp}$ of the unit-norm rotation vector $\boldsymbol{\tau}_g$ pointing in the direction of the goal. The Q-function clearly shows that the critics learn the multi-modal distribution.

Actor-critic algorithms such as **SAC** and **TD3** rely on the critic for computing the policy gradient. Hence, conflicting gradients only appear if the critic does learn the bi-modal Q-function. To test if this happens in practice, we analyze the learned Q-values between $\mathbf{q} = \pi(\mathbf{s})$ and $-\mathbf{q}$ as shown in figure 13. We sample equally-spaced actions discretely along the geodesic connecting $\mathbf{q} = \pi(\mathbf{s})$ and $-\mathbf{q}$ that passes through the goal orientation on the $S(3)$ manifold by applying the $\mathrm{Exp}$ of the unit-norm rotation vector $\boldsymbol{\tau}_g$ pointing in the direction of the goal. Both critics have learned nearly the same Q-values for $-\boldsymbol{q}$ and $\boldsymbol{q}$, although the value for the actual action $\boldsymbol{q}$ is still slightly higher in both cases. Since the double-cover points in the opposite direction in Euclidean space, the critics will produce conflicting gradients for actions sampled on the hemisphere of $-\boldsymbol{q}$.

Lastly, although not visualized here, the results directly apply to **PPO**'s advantage estimates if both action representations are sampled during the same rollout.

### A.1.7 EXTENDED RESULTS FOR ENTROPY REGULARIZATION IN **PPO** AND **SAC**

In this section, we study the effects of varying entropy levels on **PPO** and **SAC** in the idealized environment. For **PPO**, we scale entropy coefficients relative to the optimal value determined through hyperparameter tuning. **SAC**'s target entropy is scaled relative to its default value of $-dim(\mathcal{A})$.

Due to the non-Euclidean nature of rotation representations, increased entropy might not always lead to better exploration. For instance, vectors of larger magnitude in the tangent space attain

higher directional stability. Thus, from the perspective of entropy maximization, actions of large magnitude are more attractive than their smaller counterparts. This correlation between entropy and action magnitudes in the tangent space can be seen clearly in figure 14. Policies that fail to reach a zero norm oscillate near the goal indefinitely due to a lack of sufficient exploration near the identity. This effect can be mitigated by constraining the maximum rotation magnitude; hence, we always recommend scaling local tangent increments. The effectiveness of this mitigation depends on the magnitude of the maximum rotation angle $\alpha_{max}$, with smaller values being more beneficial.

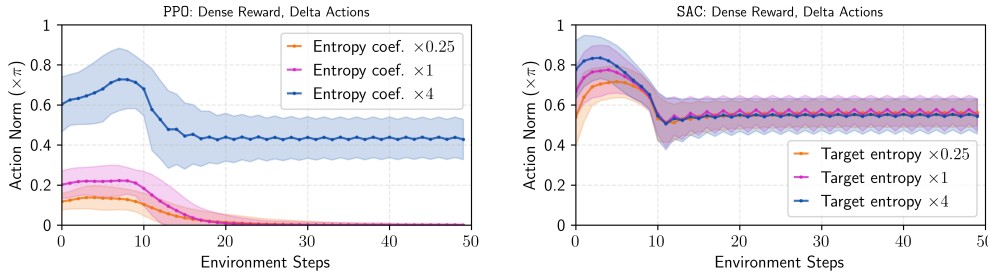

Figure 14: Action norms for different entropy levels for **PPO** (left) and **SAC** (right) using the unscaled delta tangent action representation in the idealized environment. Actions are evaluated across 3000 episodes with different goal orientations set at a magnitude of $\pi$. The agent's initial orientation is always initialized to the identity rotation.

A very similar effect manifests when using quaternions. Large imaginary components $(x, y, z)$ make the rotation direction more robust to noise perturbations. In addition, the norm of $(x, y, z)$ constrains the effect of perturbations to the real component $w$ on the rotation's magnitude. This follows from the relation that the rotation's magnitude $\theta = 2\tan^{-1}(\sqrt{x^2 + y^2 + z^2}/w)$. For **SAC**, this effect is more prominent due to its mean actions remaining off-manifold, allowing them to exceed a unit norm.

Euler angles do not suffer from the same issues. Increased entropy coefficients lead to a slight increase in action magnitudes, but this increase plateaus quickly, unlike in tangent and quaternion representations, which showcase an almost-monotonic effect. A likely explanation is that Euler angles become more non-linear as Euler angles become larger, which makes learning harder (see section A.1.2). At high entropy levels 100x above the baseline, singularities attract agents trained with Euler angle action representations. However, their performance deteriorates considerably in this parameter regime.

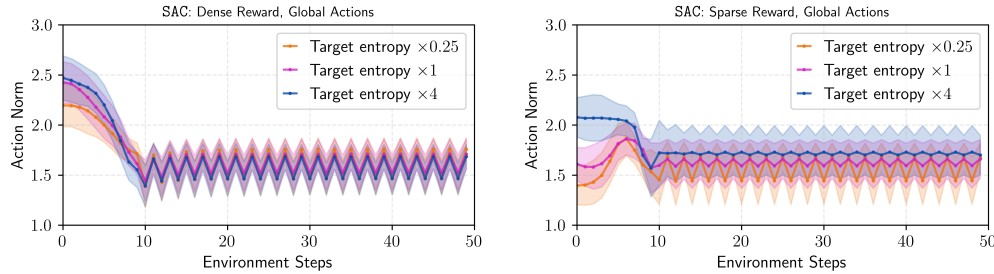

Figure 15: Action norms for different entropy levels for **SAC** across the dense (left) and sparse (right) reward settings using the global rotation matrix action representation in the idealized environment.

Lastly, rotation matrices with **PPO** are not affected by any entropy-related issues since the Frobenius norm of a 3D rotation matrix is constrained by the structure of the SO(3) manifold, such that $\|\mathbf{R}\| = \sqrt{3}$. However, due to **SAC**'s off-manifold actions, entropy maximization provides an undesired incentive for policies towards increasing the Frobenius norm of their actions. This effect, which contributes to the poor performance of matrix policies compared to **TD3**, can be seen clearly in figure 15. To explain why **SAC**'s actions remain off-manifold, refer to section A.1.3.

## A.2 Full Experimental Results on the Idealized Environment

This section completes the results presented in section 3.2. We test **PPO** with global and delta actions on dense rewards. Training goal-conditioned policies with sparse rewards makes little progress without HER; our analysis omits it. Since **SAC** and **TD3** are off-policy algorithms compatible with HER, we train them with global and delta actions on sparse and dense rewards. All experiments use 50 runs per representation to ensure the significance of the results. Hyperparameters are tuned according to section A.8.

### A.2.1 **PPO** Results

Figure 16 shows the results. Rotation matrix representations yield the best performance for global actions. Despite its lower dimensionality, the quaternion representation is less successful. We attribute this to the double-cover as established in section A.1.6. Perhaps surprisingly, tangent increments in the Lie algebra perform better than quaternions. Euler angles reach approximately the same performance as quaternions.

For delta actions, tangent vectors in the local frame display superior performance to other representations, with Euler angles as a close second. Matrix and quaternion representations struggle to learn a good policy. Reducing the randomness at the start of the training does not help these representations, because vectors centered around $0$ will still yield vastly different actions by projecting the mean onto the manifold.

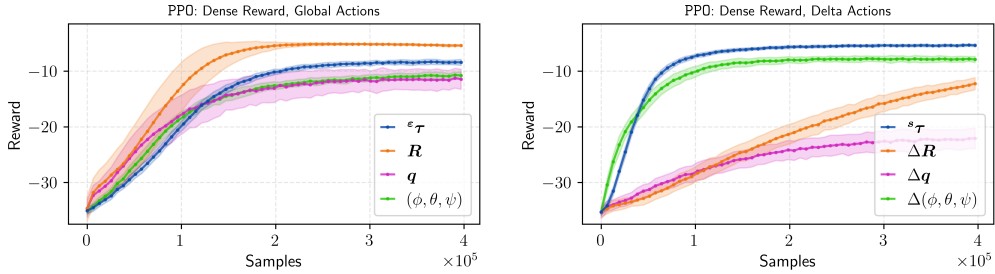

Figure 16: **PPO** learning curves for dense rewards using global (left) and delta (right) action representations in the idealized environment.

We note that the performance of the Euler representation is highly dependent on initializing the policy log-standards to a small value of $-2$. With a default value of $0$ as used in many reference implementations, agents cannot learn any reasonable policy. This trick is specific to the Euler angle representation (see section A.1.2).

We recommend the use of the delta tangent representation for **PPO**. It displays a reduced variance compared to global matrix actions, achieves a slightly better final performance, and avoids issues with projections of small actions early on during training.

### A.2.2 **SAC** Results

For dense rewards and global actions, the rotation matrix representation converges fastest. We attribute this to the uniqueness and smoothness of the representation. Quaternions are second with decreased convergence stability because of the multi-modality introduced by the double-cover of $\mathcal{S}(3)$ (see section A.1.6). For delta actions, the local tangent space representation significantly outperforms others and performs better than global actions. Attaching the tangent space to the local frame always puts the cut-locus on the farthest side from the current orientation and prevents it from impacting uniqueness and continuity. All results are shown in figures 17 and 18.

The results for sparse rewards differ significantly from the dense case. While the matrix representation remains the best for global actions, its performance is well below the optimum. The other representations improve more slowly and remain worse in performance. An analysis of the trained policies reveals that the entropy maximization in combination with the inability to project actions

(see section A.1.3) learns to maximize the entropy regularization before the critic can provide a meaningful policy gradient. Since subsequent exploration is based on the behavior of the policy and not random noise (e.g., in **TD3**), agents cannot recover from this collapse. Tangent and Euler policies are unaffected but still suffer from discontinuities and singularities.

Delta actions show a similar severe degradation in performance for matrix and quaternion representations for the same reason as in the global case. Tangent spaces are unaffected and achieve near-optimal performance. Delta Euler angles outperform the matrix and quaternion representations, but remain significantly below the tangent representation and experience the largest variation between training runs (see section A.1.4).

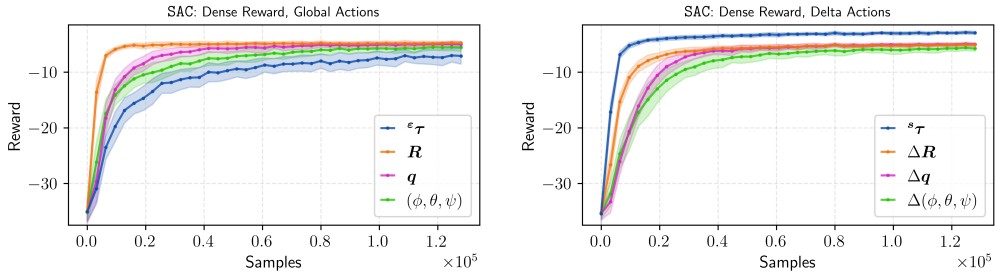

Figure 17: **SAC** learning curves for dense rewards using global (left) and delta (right) action representations in the idealized environment.

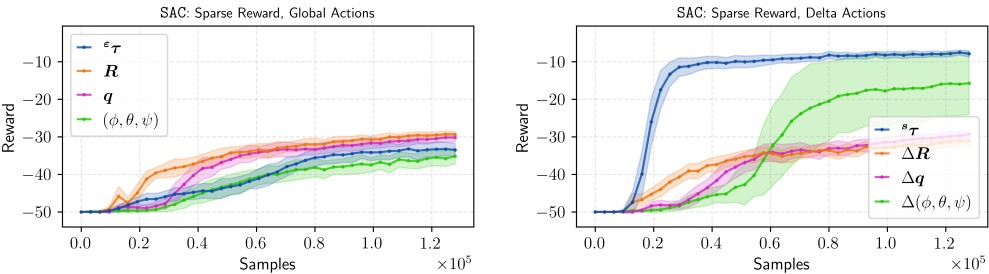

Figure 18: **SAC** learning curves for sparse rewards using global (left) and delta (right) action representations in the idealized environment.

As with **PPO**, we recommend using the delta tangent space representation with **SAC**. If global action spaces are required, practitioners should opt for a matrix or quaternion representation, but have to ensure that dense rewards are available to prevent agents from collapsing into degenerate action outputs.

### A.2.3 **TD3** RESULTS

As in **SAC**, the rotation matrix representation in **TD3** converges fastest and displays the highest performance for dense rewards and global actions in figure 19. Quaternions are second again, for the same reasons as for **SAC**. Again, the local tangent space representation significantly outperforms all others for delta actions, and achieves superior performance compared to global actions.

The results in figure 20 for sparse rewards show the same results, but amplify the differences between representations. The matrix representation has a more apparent advantage in the global frame, while Euler and the Lie algebra representations struggle to learn a successful policy. In the local frame, the tangent representation converges almost immediately to the optimal policy. On the other hand, matrix, quaternion, and Euler angle representations display large variances between training runs and are often unable to find a successful policy.

As in **PPO** and **SAC**, our recommendation for **TD3** is to use the delta tangent space representation. If global action spaces are required, practitioners should opt for a matrix or quaternion representa-

tion. Contrary to `SAC`, exploration is less of an issue with these representations due to the policy-independent random exploration noise. Hence, matrix and quaternion representations also work with sparse rewards.

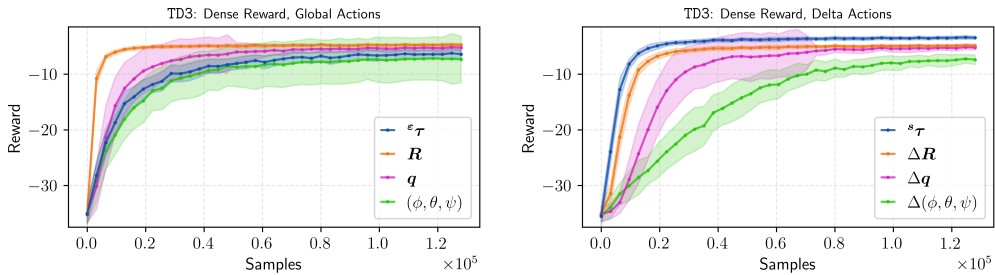

Figure 19: **TD3** learning curves for dense rewards using global (left) and delta (right) action representations in the idealized environment.

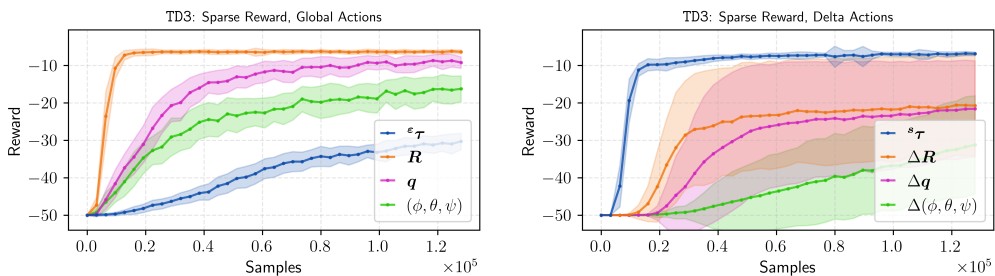

Figure 20: **TD3** learning curves for sparse rewards using global (left) and delta (right) action representations in the idealized environment.

### A.3 EXTENDED DISCUSSION OF UNIT ROTATION-CENTERED DELTA ACTIONS

Section 2.4.1 discusses that policy networks using quaternion and matrix representations are not centered around the unit rotation for delta actions, potentially leading to worse performance. To address this, the network output can be modified such that a zero output corresponds to the identity rotation $\boldsymbol{I}$ (or unit quaternion $\boldsymbol{q_I}$). We investigate two distinct implementations of this centering strategy for quaternion and matrix representations and evaluate their impact on learning stability and final performance.

#### A.3.1 COMPARISONS OF IMPLEMENTATIONS

We denote the raw output of the neural network as $\boldsymbol{q}_{\text{net}}$ and $\boldsymbol{R}_{\text{net}}$, respectively. We define two variants for mapping this output to a pre-projection action $\boldsymbol{q}_{\text{pre}}$ and $\boldsymbol{R}_{\text{pre}}$, which are subsequently projected to obtain the valid rotation.

**Additive Bias.** The simplest approach is to add the identity element to the network output.

$$\boldsymbol{q}_{\text{pre}} = \boldsymbol{q}_{\text{net}} + \boldsymbol{q_I}$$
$$\boldsymbol{R}_{\text{pre}} = \boldsymbol{R}_{\text{net}} + \boldsymbol{I}$$

This variant ensures that if $\boldsymbol{q}_{\text{net}}$ or $\boldsymbol{R}_{\text{net}} \approx 0$, the resulting action is close to the identity. However, this method limits the set of reachable actions. Assuming $\tanh$ activations bound $a_{\text{net}}$ within $[-1, 1]$, the scalar component of $\boldsymbol{q}_{\text{pre}}$ is restricted to the range $[0, 2]$. Consequently, the policy cannot represent the conjugate unit quaternion $\boldsymbol{q_I^*}$ directly. A similar property holds for the matrix representation.

**Scaled Bias.**  To mitigate the range limitation of the additive variant, we can also implement the mapping with a scaling factor dependent on the identity element.

$$q_{\text{pre}} = q_{\text{net}} \odot (q_I + 1) + q_I$$
$$R_{\text{pre}} = R_{\text{net}} \odot (R_I + 1) + I$$

Here, $\odot$ denotes element-wise multiplication and $1$ is a tensor of ones with matching dimensions. By scaling the network output, the policy can reach all possible quaternion configurations, including the negative hemisphere. However, this introduces non-uniform sensitivities, since the scalar component $w$ and the diagonal elements of the rotation matrix, respectively, induce larger changes in the resulting rotation magnitude compared to the additive variant.

### A.3.2   RESULTS ON THE IDEALIZED ENVIRONMENT

We benchmark both implementations against the standard, uncentered parameterizations in our idealized environment. Results are averaged over 50 runs as in the main text.

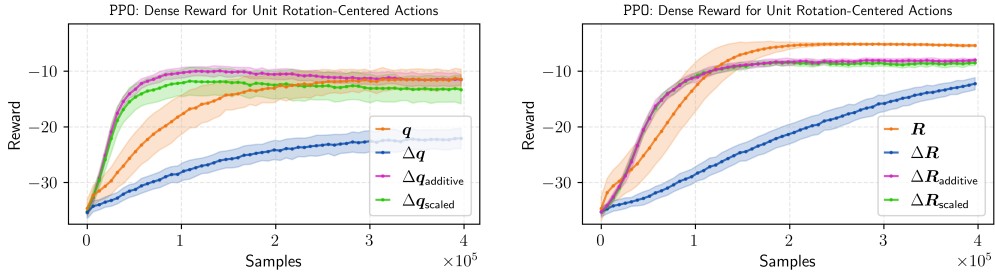

Figure 21: **PPO** learning curves for unit rotation-centered delta actions using delta quaternion (left) and matrix (right) action representations in the idealized environment.

For **PPO**, the modified relative actions provide a clear benefit. Both converge slightly faster than their absolute variants and significantly improve over the uncentered delta actions (see figure 21). Comparing the two variants, the additive mapping achieves a slight performance advantage over the scaled one. For quaternions, only the additive variant achieves similar performance to the absolute representation. For the matrix representation, both variants converge to a lower performance than the absolute representation.

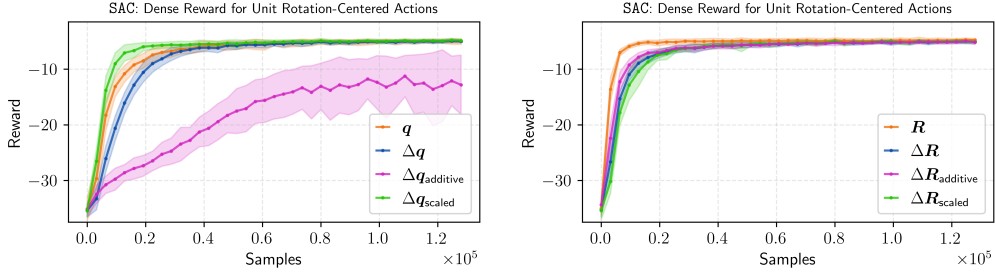

Figure 22: **SAC** learning curves for unit rotation-centered delta actions and dense rewards using delta quaternion (left) and matrix (right) action representations in the idealized environment.

The results for **SAC** present a contrast to **PPO**. As can be seen in figure 22, in dense rewards, the scaled variant slightly improves over regular delta and absolute actions for quaternions, whereas the additive one leads to significantly worse performance. The matrix representations are almost unchanged compared to the default implementation. In sparse rewards (see figure 23), the additive version performs slightly worse compared to the default delta orientation, whereas the scaled version slightly outperforms it. None of the three, however, yields a successful policy. The same is true for the rotation representation.

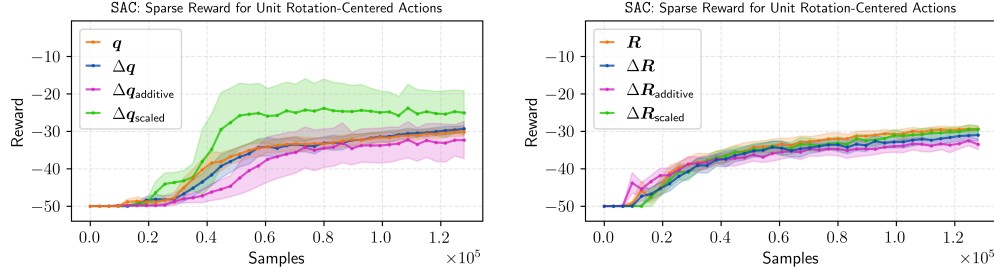

Figure 23: **SAC** learning curves for unit rotation-centered delta actions and sparse rewards using delta quaternion (left) and matrix (right) action representations in the idealized environment.

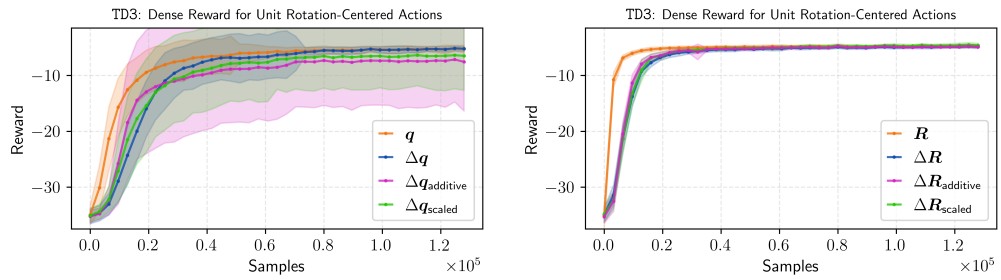

Figure 24: **TD3** learning curves for unit rotation-centered delta actions and dense rewards using delta quaternion (left) and matrix (right) action representations in the idealized environment.

**TD3** shows a similarly mixed picture in figure 24, where both variants slightly underperform with dense rewards compared to the default quaternion implementation, and have significantly increased variances between runs. The matrix representation shows no real difference between the default implementation and the improved versions. In sparse rewards, both modified quaternion representations outperform the baseline, but still exhibit significant variance and fall short of the consistency and performance of the absolute representation. The modified matrix representations in figure 25 significantly outperform the baseline and approach the performance of the absolute representation, but have a higher variance between runs.

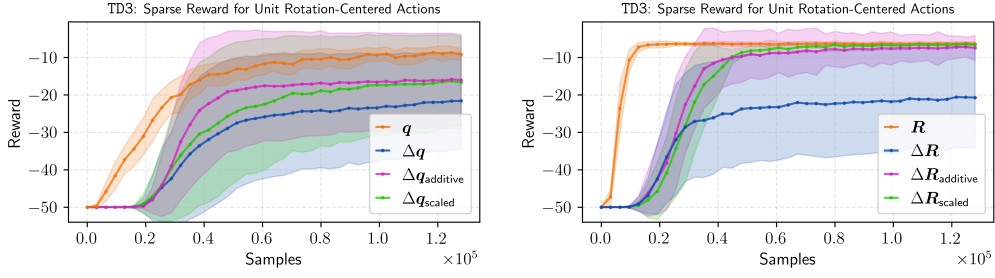

Figure 25: **TD3** learning curves for unit rotation-centered delta actions and sparse rewards using delta quaternion (left) and matrix (right) action representations in the idealized environment.

Our experiments indicate that while unit-centering addresses some of the issues associated with delta rotations, its practical impact varies by algorithm. The performance difference is negligible for many configurations. The notable exceptions are **PPO**, where centering is crucial for competitive performance of delta actions, and **SAC**, where the scaled variant is favorable for quaternions. In the specific case of **PPO**, we observe a slight advantage of the additive variant over the scaled one.

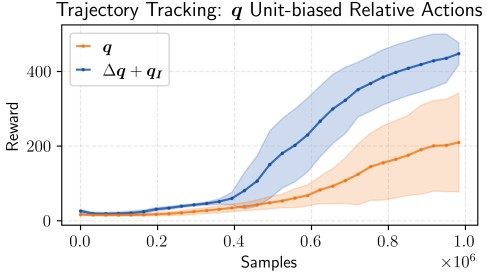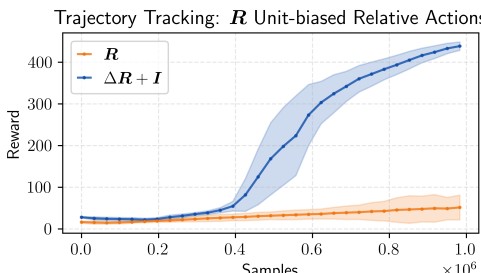

Figure 26: **PPO** learning curves for the drone trajectory tracking environment using delta quaternion (left) and matrix (right) action representations with unit rotation bias.

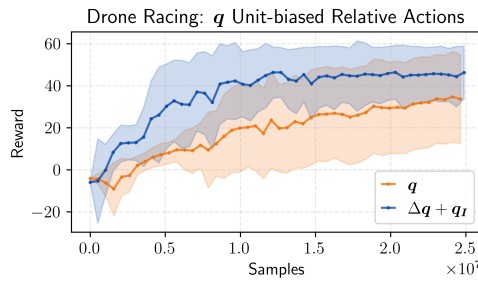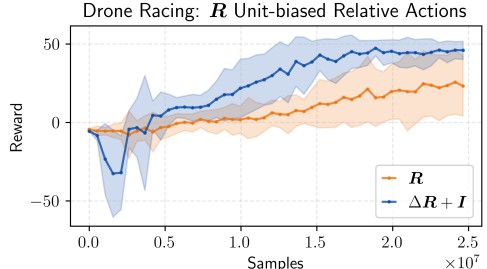

Figure 27: **PPO** learning curves for the drone racing environment using delta quaternion (left) and matrix (right) action representations with unit rotation bias.

**Revisiting Unstable Systems with Unit-biased Relative Actions.** In section 4, we hypothesized that zero-centered actions are particularly beneficial for unstable robotic systems, such as drones, where maintaining the initial attitude is safer than a random rotation. Centering actions around the unit rotation should theoretically allow quaternion and matrix representations to stabilize and converge faster on these tasks.

Since the additive variant performed best for **PPO**, we selected it to validate this hypothesis on the trajectory tracking and drone racing benchmarks. We compare this centered variant against the global action baseline for both quaternion and matrix actions. For a fair comparison, we retune the hyperparameters using the same sweep settings as for the baseline.

Our results confirm the hypothesis. The modified relative quaternion actions in figure 26 demonstrate dramatically improved convergence speed compared to uncentered variants. Similar holds true for matrices, although neither manage to outperform the local tangent representation. By initializing the policy output effectively at the identity rotation, the drone is less prone to erratic initial behavior that leads to immediate crashes, thereby allowing the agent to gather meaningful experience earlier in the training process.

While less dramatic, our results on the drone racing task in figure 27 clearly show the same trend. We conclude that for unstable systems, the unit-centering of representations is a key influence on performance.

### A.4 COMPARISON OF ROTATION MATRIX AND 6D REPRESENTATIONS

Beyond the standard parameterizations discussed in the main text, several previous studies have explored suitable rotation representations for learning. One influential representation, particularly noted for its properties in optimization and supervised learning contexts, is the continuous 6D representation proposed by Zhou et al. (2019).

The idea behind this representation is that a full 9-element rotation matrix is over-parameterized. Defining two orthogonal column vectors is sufficient to reconstruct the complete orientation without

losing the uniqueness and smoothness of the representation. Since a rotation matrix belonging to SO(3) describes a right-handed coordinate system, the third axis is entirely defined by the cross product of the first two. Implementation-wise, the policy outputs a 6D vector $x \in \mathbb{R}^6$ that is treated as two raw 3D vectors. To ensure orthogonality and unit norm, we then apply the Gram-Schmidt process to these vectors.

While this representation enjoys a lower dimensionality than the standard flattened rotation matrix, Geist et al. (2024) point out that it empirically falls short of the standard SVD-projected matrix representation on several tasks. Since the arguments regarding continuity and smoothness apply similarly to both the 6D representation and standard matrices, and given that rotation matrices are more prevalent in standard robotics pipelines, we selected matrices for our primary comparison. In this section, we provide a direct comparison to demonstrate that the performance of the 6D representation is indeed comparable to, but rarely strictly better than, the standard matrix representation in reinforcement learning settings.

### A.4.1 RESULTS ON THE IDEALIZED ENVIRONMENT

We benchmark the 6D representation against the standard matrix representation across all algorithms with 50 runs per variant as in the main text.

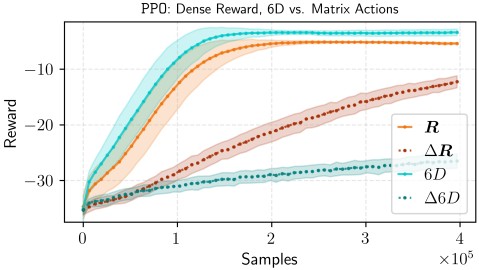

Figure 28: **PPO** learning comparison between the 6D representation and rotation matrices in the idealized environment for both absolute (straight lines) and delta (dotted lines) actions.

For **PPO**, the 6D representation slightly outperforms the standard matrix representation in terms of both convergence speed and final performance when using absolute actions. Conversely, for delta representations, the 6D parameterization falls short of the matrix baseline (see figure 28).

The results for **SAC**, visualized in figure 29, present a similar picture. In the dense reward setting, the performance of relative and absolute representations is practically indistinguishable between the two parameterizations. In the sparse reward setting, we observe a minor improvement for the 6D representation across both absolute and relative modes. However, it is important to note that neither

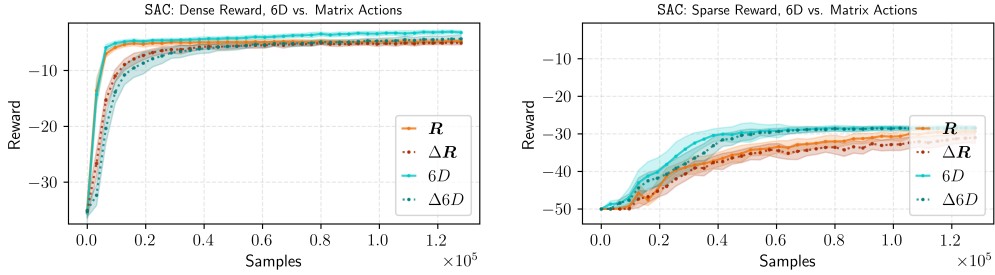

Figure 29: **SAC** learning comparison between the 6D representation and rotation matrices in the idealized environment for absolute (straight lines) and delta (dotted lines) actions on dense (left) and sparse (right) rewards.

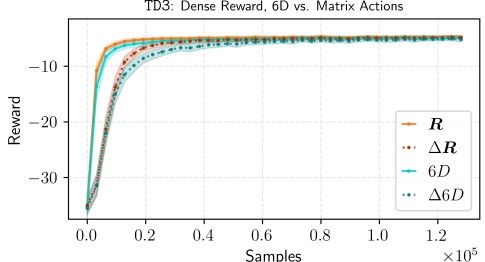
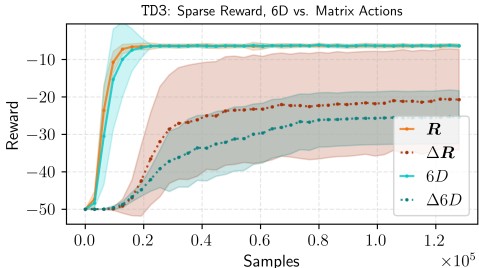

Figure 30: **TD3** learning comparison between the 6D representation and rotation matrices in the idealized environment for absolute (straight lines) and delta (dotted lines) actions on dense (left) and sparse (right) rewards.

the matrix nor the 6D representation successfully learns a high-quality policy under these sparse reward conditions, sharing the same failure modes and training characteristics.

In **TD3**, the standard matrix variants slightly outperform their 6D counterparts generally. The only meaningful difference exists between the delta matrix and the delta 6D actions under sparse rewards, where the matrix representation converges to a better policy on average (see Figure 30). However, consistent with our main results, both representations yield relatively poor policies in this setting.

Overall, the performance curves of matrix and 6D representations are very similar across runs. **PPO** with absolute actions shows a slight preference for the 6D representation, whereas **TD3** generally favors the standard matrix representation. A possible explanation for this result is the increased robustness of the full matrix representation to large disturbances such as the uniform exploration noise used in **TD3**, compared to the Gram-Schmidt reconstruction used in the 6D mapping. However, we emphasize that these performance differences are relatively minor and should not be overinterpreted as a definitive advantage for either representation.

**Robot Benchmarks for PPO**    Since the 6D representation slightly outperformed its matrix counterpart on **PPO** (see figure 28), we rerun the robot benchmarks that use **PPO** for trajectory following and drone racing.

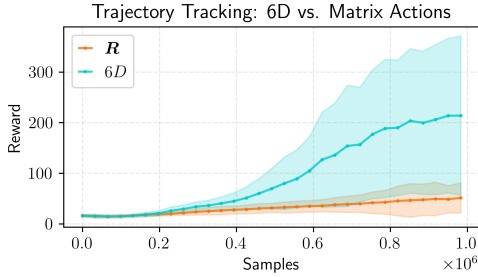
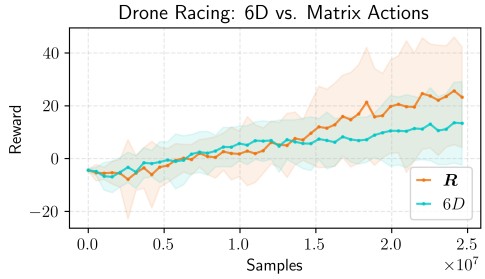

Figure 31: **PPO** learning comparison between the 6D representation and rotation matrices on the trajectory following task (left) and on drone racing (right).

The results in figure 31 display an inconclusive picture. While the 6D representation improves notably on the trajectory following task, the deviations between runs are large. Other effects such as unit-centering of delta actions (see section A.3) are significantly more consistent in improving convergence. The results on racing are again in line with our intuition that the two representations perform very similar. Although matrix marginally outperforms the 6D representation, the difference is not significant.

### A.5 IMPLEMENTATIONS

Even when the underlying concepts are clear, putting SO(3)-specific operations into code can be difficult. We explicitly outline some of the important operations mentioned in the paper to help practitioners transfer the results.

We detail how we rotate around tangent vectors ${}^s\tau$ in the body frame, implement exponential and logarithm maps for rotation scalings, and perform differentiable matrix orthonormalization inside neural network layers. All components are implemented using standard scientific Python libraries only. Note that there is ongoing work within SciPy (Virtanen et al., 2020) toward rotation routines compatible with most deep learning frameworks, further simplifying implementations (Schuck et al., 2025b).

Listing 1: Rotate an orientation by a local tangent vector

```python
import numpy as np
from scipy.spatial.transform import Rotation as R

current_orientation = R.identity()
# Action in [-1, 1]^3 from the policy network
action = np.random.default_rng().uniform(-1, 1, size=3)
# Apply the tangent vector to the current orientation as delta action
next_orientation = current_orientation * R.from_rotvec(action)
```

Listing 2: Limiting rotations to a maximum angular change

```python
import numpy as np
from scipy.spatial.transform import Rotation as R

current_orientation = R.identity()
# Limit the rotation to a maximum of pi/10 radians
max_rotation = np.pi / 10
# Calculate the next orientation in the direction of the global reference
action = np.array([0, 1, 0, 0])  # Example quaternion action
# Compute the difference between current and target orientation
delta = current_orientation.inv() * R.from_quat(action)
# Check if the rotation angle exceeds the maximum allowed. If so, scale
# the magnitude to at most the maximum allowed rotation
scale = np.minimum(1, max_rotation / delta.magnitude())
# Apply the scaled delta to the current orientation. Power on rotation is
# equivalent to log -> scale -> exp
next_orientation = current_orientation * delta**scale
```

Listing 3: Differentiable matrix orthonormalization in PyTorch

```python
import torch

def orthonormalization(x: torch.Tensor) -> torch.Tensor:
    u, _, vh = torch.linalg.svd(x)
    # Ensure det = 1 without in-place changes to u (breaks backprop)
    unorm = torch.zeros_like(u)
    unorm[..., :2] = u[..., :2]
    unorm[..., 2] = u[..., 2] * torch.det(u @ vh)
    return unorm @ vh

action = torch.randn(3, 3)
norm_action = orthonormalization(action)  # Orthogonal, det 1
```

A.6 ENTROPY ON THE MANIFOLD

The standard entropy bonus term employed in **PPO** and **SAC** is

$$\mathcal{H}(\pi_\theta(\cdot|s)) = \frac{1}{2}\sum_{i=1}^{4}\log\left(2\pi e \sigma_i^2(s)\right). \tag{5}$$

A policy that parametrizes rotation actions as quaternions with zero norm and unit variance on all action dimensions will lead to a uniform distribution on the $\mathcal{S}(3)$ sphere after normalization and thus a uniform distribution of rotations with maximum entropy. If we shift the mean of $q_w$ towards 1, the entropy in equation 5 remains constant, while the actual entropy of the rotation distribution declines. The decline is evident because the entropy of the distribution on the sphere

$$\mathcal{H}[X] = -\int_{\mathcal{S}(3)} p(x)\log p(x)d\Omega(x) \tag{6}$$

with $d\Omega$ as the surface area measure and a random variable $X \in \mathcal{S}(3)$ has its maximum at $p(x) = \frac{1}{2\pi^2}$. One can show this is the global maximum with a variational argument where the entropy functional is concave in $p$.

The implication for training RL agents is that agents can increase their mean to higher values to concentrate the distribution of rotations in $SO(3)$ while still receiving the same entropy bonus as uniformly distributed rotations, thus effectively breaking entropy regularization.

A.7 FETCH ORIENTATION ENVIRONMENTS

The `ReachOrient` and `PickAndPlaceOrient` environments (see figure 32) used in section 4 to benchmark **TD3** are directly inspired by the `Fetch` environments (Andrychowicz et al., 2017). However, to make them compatible with the extended scope of our environments, we made some modifications, which are listed below.

The most fundamental change is the extended action space. In addition to the arm's position and finger joints, agents also control the arm's orientation. The maximum angular change of the arm between two environment steps is $\frac{\pi}{10}$ radians, as in our idealized rotation control environments.

A goal orientation has extended the environment's goal. In the case of `ReachOrient`, this target orientation is sampled by randomly rotating the arm start orientation by at most $\frac{\pi}{2}$ radians. We limit the goal orientation to prevent infeasible configurations that are physically infeasible for the arm. `PickAndPlaceOrient` samples its orientation goal similarly with respect to the cube orientation. As in the original environment, half of the goal positions are sampled uniformly above the table, and half are sampled on the table itself. Goal orientations on the table are sampled to be physically feasible. The cube is randomly rotated at the start of each episode by sampling which side is facing down and then rotating it uniformly around its $z$-axis.

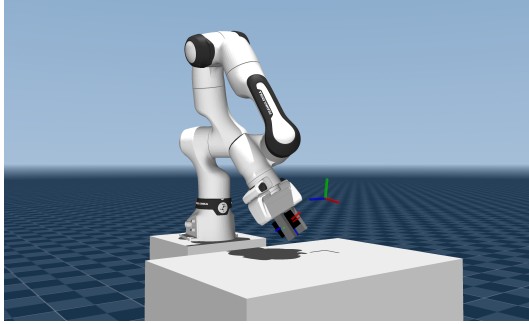

Figure 32: Example rollout of the `PickAndPlaceOrient` environment. The agent has to pick up the cube and place it into the same position and orientation as the goal frame. The frame located to the right of the robot arm indicates the goal pose.

Changes in the goal formulation also require a change to the reward function. Agents receive a reward of 0 if the position and orientation of the arm (`ReachOrient`) or cube

Table 3: Number of runs per environment for hyperparameter optimization

| Environment | Number of Runs |
|---|---|
| Idealized Rotation Environments | 100 |
| Drone Trajectory Tracking | 100 |
| Drone Racing | 50 |

(`PickAndPlaceOrient`) are both within their respective tolerances of $5$cm and $\frac{\pi}{10}$ radians, and $-1$ everywhere else.

Andrychowicz et al. (2017) do not consider orientation control, and thus it has no influence on the choice of robot. However, the Fetch robot shows a limited range of reachable position and orientation targets. We thus exchange the robot with the Franka FR3, one of the most common robot arms in robotics research. The gripper remains from the original environments to limit implementation variations.

## A.8    Hyperparameter Choices

The hyperparameters used for each algorithm and action representation are optimized per environment using Bayesian optimization (Bergstra et al., 2011). Table 3 lists the number of runs used for hyperparameter tuning across each environment.

We determine separate sets of hyperparameters for the idealized rotation environment on the sparse and dense reward settings. However, delta and global actions of the same representation share their hyperparameters. The action viewpoint does not significantly change the choice of hyperparameters in trial runs. Across the `Fetch` environments, we reuse the same hyperparameters used by Andrychowicz et al. (2017) to maintain consistency with prior results. Similarly, for the RoboSuite (Zhu et al., 2020) benchmark, we adopt the same original hyperparameters used for **SAC**.

All our experiments use identical MLPs for both policy and value networks. For **PPO**, we use 2-layer networks with 64 units per layer and `tanh` activations. The only exception is drone racing, where we find that using 128 units per layer, as per (Kaufmann et al., 2023), is beneficial. Note that similar increases in the number of units per layer across other tasks with **PPO** negatively affect performance. Both **SAC** and **TD3** use 3-layer networks with 256 units per layer and ReLU activations. However, for **SAC**'s RoboSuite benchmark, we use 2-layer networks instead, following the architecture originally used in the benchmark for full pose control.

## A.9    Usage of Large Language Models

We used large language models (LLMs) for wording and proofreading individual sections. In addition, they were used as a programming aid to create the TikZ figures. The ideation phase made no use of LLMs.

