# OpenReview forum: "A Primer on SO(3) Action Representations in Deep Reinforcement Learning"
_ICLR.cc/2026/Conference — ICLR 2026 Poster_

### Official Review · Reviewer_gmAW · 2025-10-28

**Soundness:** 3
**Presentation:** 2
**Contribution:** 2
**Rating:** 4
**Confidence:** 4

**Summary:**

Authors investigated the importance of 3D rotation representations in the context of reinforcement learning. Authors further discussed 4 rotation representations, i.e., rotation matrix, quaternion, Euler angle, and Lie algebra, in absolute and relative coordinates. Extensive simulation experiments are performed on drone control as well as manipulation problems. Authors concluded that on average, the relative Lie algebra is the best performant representation for RL applications. To explain the advantage of relative Lie algebra, various hypothesis are presented.

**Strengths:**

Despite previous studies on rotation representations in the context of supervised learning, this paper explores rotation representations in the RL setup, which is an important topic but is ignored by previous works. This paper provided extensive comparisons on different RL benchmarks, ranging from drop control to manipulation. This paper also compared different RL algorithms, including stochastic and deterministic algorithms. The hypothesis point interesting explanation of the advantages of relative representations.

**Weaknesses:**

One important baseline seems messing: 6D rotation representation [1], where the neural network first outputs two 3D vectors, then uses Gram-Schmidt to normalize these vectors and thus reconstructs the rotation matrix. How would this baseline, and perhaps it’s relative version, performs in the RL tasks?

[1] On the Continuity of Rotation Representations in Neural Networks. Yi Zhou, Connelly Barnes, Jingwan Lu, Jimei Yang, Hao Li.

**Questions:**

Table 1 is very helpful for explaining different rotation representations. Could authors add similar table to summarize the representations on Table 2? E.g., R stands for rotation matrix, \delta R stands for relative rotation matrix.

How does Figure 2 visualize SO(3) rotation? Figure 2 seems visualized a S2 sphere, which only has 2 DoFs, but SO(3) has 3 DoFs.

Why Figure 3 only compared relative rotation representations? What do absolution representations perform?

For Figure 4, an additional bar that shows the average of each representation would be helpful for overall comparison.

---

> ### Author Response · Authors · 2025-11-19
> **Response to Reviewer gmAW**
>
> Dear reviewer,
>
> We would like to thank you for your review and the points you raised, and we hope to clarify them in our response.
>
> ## Missing baseline
> As correctly pointed out, the baseline from Zhou et. al. [1] is missing. We made this decision to limit the number of representations so that we could focus more on the effects of individual representations. We reasoned that the proposed 6D representation was sufficiently similar in character to rotation matrices, so we only considered the more widely used matrix representation. In addition, Geist et al. [2] reported that the Gram-Schmidt algorithm leads to unbalanced sensitivities, which sometimes results in worse performance. They recommend using the matrix representation, which is why we made our decision.
>
> Nevertheless, it is an important baseline, and might be of greater interest to the community than we anticipated, given your review. We have thus implemented the baseline and included a section in the appendix that focuses solely on the comparison between the matrix and 6D representation.
>
> The results are included in Appendix A.4. We find that the 6D representation slightly improves the final performance for the absolute matrix representation in PPO, but significantly underperforms for relative actions (see Fig. 26). The only other meaningful difference is in TD3 under sparse rewards, where the delta 6D representation performs weaker than the matrix counterpart (see Figure 28). In general, the training performance of the representations roughly matches, which confirms our initial reasoning.
>
> Since absolute actions using the 6D representation in PPO slightly improved performance, we are currently running additional experiments to validate whether this also holds for the robotic tasks. We require extra time to complete the sweeps for the trajectory tracking and drone racing tasks, but we will add the results to the section once they are ready.
>
> Unfortunately, the space constraints of the paper made it impossible to include the 6D representation in the main paper. We hope you find the dedicated comparison and extensive runs are sufficient to cover the topic thoroughly.
>
> In summary, we have made the following changes to the paper for the 6D representation:
> - Added a note to section 3.2 explicitly referencing Zhou et al. and referencing the discussion in the appendix
> - Added Appendix A.4 with an overview of the 6D representation, a brief explanation of our thought process, and the complete set of experiments on the idealized environment.
> - Will add the benchmarks for drone control once they have finished.
>
> ## Questions
>
> **Adding another table for the representations:**
>
> There would be a significant overlap between Table 1 and the potential table on action representations, but we do agree that collecting the notation in a table is helpful. We thus modified Table 1 to include the symbols for relative actions.
>
> **How does Figure 2 visualize SO(3) rotation?**
>
> Figure 2 indeed shows a sphere, but the samples do not have to lie on the surface of the S(2) sphere; they can also lie within it and thus we have 3 DoFs. We considered how to best visualize 3D orientations in our plots, and the tangent space at the origin seemed the most reasonable choice for us. We have adjusted the figure description to clarify this and improved the visibility of the axes.
>
> **Why are only relative rotations shown in Figure 3?**
>
> Figure 3 shows the relative rotations for the tangent space and Euler angles, as we found these to perform better than their absolute counterparts in Section 3. For rotation matrix and quaternion representations, it shows the performance of using absolute actions, again, because we found them to perform better than their relative counterparts. We remark this at the beginning of section 4. Running all benchmarks for all combinations, especially for the Robosuite benchmark, exceeded our compute budget. We hope the distinction of relative and absolute actions is more apparent now with the addition to Table 1 you suggested.
>
> **Mean performance in Figure 4:**
>
> Figure 4 now also displays the mean reward over all tasks under “Average over Tasks” as per your suggestion.
>
> ## Summary
>
> We hope that we have addressed all your concerns on the baseline and the questions to your satisfaction. Please let us know if you have any further comments on our work or the steps we took in response to your review. If our clarifications address your concerns, we would appreciate it if you would consider this in your evaluation.
>
> ## References
> [1] Yi Zhou, Connelly Barnes, Lu Jingwan, Yang Jimei, and Li Hao. On the Continuity of Rotation Representations in Neural Networks. In The IEEE Conference on Computer Vision and Pattern Recognition (CVPR), June 2019.
> [2] A. Ren´e Geist, Jonas Frey, Mikel Zhobro, Anna Levina, and Georg Martius. Learning with 3D rotations: a Hitchhiker’s guide to SO(3). In Proceedings of the 41st International Conference on Machine Learning, 2024.

---

> ### Author Response · Authors · 2025-11-22
> **Text highlighting**
>
> We have now highlighted changes we made in the paper in red to make them easily visible to reviewers.

---

> ### Author Response · Authors · 2025-11-27
> **Completion of experiments**
>
> Dear reviewer,
>
> following your review, we made multiple changes to our paper. As outlined in our previous comment, we added a full discussion of the 6D representation by Zhou et al. [1] in A.4. We also
> - included the representation notation into table 1
> - clarified the description of figure 2
> - made the labels of figure 3 more explicit (since they are now added prominently in table 1)
> - added the additional bar requested for figure 4
>
> ### Results on the 6D vs Matrix comparison
> We discussed the results in our previous comment already. We now also have results on the drone racing tasks, since R6 slightly improved over matrices for PPO. As with the idealized environments, we see mixed results. R6 does improve over matrix representations on the trajectory tracking task, but the variance between runs is large, and other factors such as unit-centering of rotations (see section 2.4.1 and A.3) are significantly more impactful. On drone racing, 6D performs slightly worse than matrix, but the difference is not significant. We conclude that R6 has mostly comparable performances to the matrix representation, and does not display fundamentally different effects in our scenarios.
>
> As previously mentioned, all changes to the paper are marked in red for your convenience. We would be very grateful if you could let us know if your criticism of the missing baseline has been thoroughly addressed and your questions have been answered to your satisfaction. If so, we would appreciate it if you considered this in your score. Otherwise, please let us know what you are missing in the current, improved version of the paper.
>
> [1] On the Continuity of Rotation Representations in Neural Networks. Yi Zhou, Connelly Barnes, Jingwan Lu, Jimei Yang, Hao Li.

---

### Official Review · Reviewer_a6NH · 2025-11-01

**Soundness:** 2
**Presentation:** 3
**Contribution:** 4
**Rating:** 6
**Confidence:** 5

**Summary:**

The paper analysis how SO(3) rotation actions should be parameterized to facilitate reinforcement learning using PPO, SAC, and TD3. In particular, the paper analyzes how using the changes in rotation as action representation could benefit learning. First the paper discusses the different rotation representations and how to predict changes in rotation from deterministic / stochastic policy networks. In a simply toy example, the authors show that indeed the different rotation representation affect learning in particular for sparse reward settings. Afterwards, the authors discuss different hypothesis on why certain rotation representations show better performance. In particular, the authors consider the effect of smoothness on learning, interplay of rotation projections and exploration / entropy regularization, and action scaling. Finally, the paper provides extensive experimental benchmarks that underline the importance of rotation representations for reinforcement learning in SO(3).

**Strengths:**

The paper looks at a critical issue for the reinforcement learning community that so far has been mostly overlooked. The paper is well structured, clearly written, and the experiments are extensive and thorough. Considering how rotation projection affects exploration in RL is quite clever. I generally enjoyed reading the paper and consider it an excellent fit for ICLR.

**Weaknesses:**

**Comment 1: Each delta rotation representation requires special treatment to perform well.**

In Section 2.4, the authors outline how the Euclidean network outputs are mapped onto the rotation manifolds. For deterministic networks, the action are simply projected onto the respective manifold via a differentiable projection layer. For stochastic policies (as far as I can guess from the very brief description), the network mean is projected onto a rotation manifold, then points are sampled around this projected vector in Euclidean space, and these points are also projected onto the rotation manifold. **These approaches on obtaining rotations from deterministic/stochastic policies  are the basis of all of the subsequent discussion and therefore quite important.**

The problem is that neural networks are designed around zero-centered activations which with standard parameter initialization output zero-centered output distributions. This becomes a problem, if we want to fairly compare rotation action representation using rotation changes. The **unit rotation for lie algebra rotations and Euler angles is a zero-vector**. However, the unit rotation for quaternions and rotation matrices equals one or several unit vectors. In turn, if a network has to directly predict changes in quaternions or rotation matrices, then it has to first learn the unit rotation operation while for lie algebra rotations this is clearly not the case. In the paper (Table 2), the authors conclude that using changes in rotation is only sensible for the Lie Algebra rotation and Euler angles while predicting changes of quaternions and rotation matrices shows drastic drops in performance. I bet this is because of the aforementioned difference in the representation's respective unit rotation vectors.

I also suspect that Lie Algebra changes in rotation could be particularly well suited for modelling rotation actions. However, if setup correctly, I am positive that modeling changes in rotation via quaternions and rotation matrices would not show such drastic decreases in performance as reported by the authors if setup "correctly". As the author's paper acts as a primer for the RL community, not carefully evaluating how changes in rotations need to be parameterized could lead to rushed conclusions.

To provide a fair comparison between different representations of changes in rotation, I suggest you **model changes in quaternions / rotation matrices by adding the network output to the unit  rotation and then project the resulting vector onto the rotation manifold**. For stochastic policies, you also could add the network mean to the unit rotation and then sample in Euclidean space around this mean.

The authors have now the following courses of action:

1) Make a convincing argument why my above assessment is flawed (e.g. you did exactly the approach I have described above). If convincing, I increase my rating to an accept.
2) Extend the discussion and experiments in Section 2 and 3 to consider the above proposed approach to predicting changes in Quaternions / Rotation matrices. If done thoroughly, I increase my rating to a strong accept.

**Questions:**

Please see Comment 1 above.

---

> ### Author Response · Authors · 2025-11-19
> **Response to Reviewer a6NH**
>
> Dear reviewer,
>
> Thank you for the review, for considering this an excellent fit, and especially for being so clear about what we can improve.
>
> ## Unit-rotation centered actions
> Your primary concern is about the relative quaternion and matrix representations, namely that they are not centered around the unit rotation and might perform better if the initial output of the network is close to the unit rotation. Incidentally, we debated whether we should use this representation by default during our work on the paper. In the end, we decided against it because most practitioners will use existing frameworks without much customization of the output heads. Our primary concern in the paper was to provide straightforward guidelines for this case. However, we do agree that a thorough treatment of the topic should cover this modification.
>
> Following your review and earlier discussions, we have identified two options for implementing the delta rotations.
> - We can leave the network output range as is ($[-1, 1]$) and add the unit rotation to the output. The benefit is that the action dimension has a uniform impact on the Euclidean vector. The drawback is that some quaternions and matrices are no longer representable (e.g., the $[0, 0, 0, -1]^T$ quaternion)
> - We can use a modified version of your proposed solution that can still represent all quaternions and matrices by scaling a weighted vector, i.e., for quaternions:
> ```python
> q_ident = torch.tensor([0, 0, 0, 1.0])
> torch.clamp((torch.tanh(x) * (q_ident + 1)) + q_ident, min=-1, max=1)
> ```
> and similar for matrices. We ran the complete set of algorithms on the idealized environment for both implementations and found that both are similar in performance.
>
> Our results support your hypothesis for PPO, where the modifications indeed improve performance (see Figure 21 in the updated paper). However, the unit-centered delta networks still do not outperform their absolute counterparts on most algorithms, and are sometimes detrimental (see SAC in Figure 22). We also find that they do not perform better than tangent vectors.
>
> We still think that zero-centered actions have an impact in environments where the orientation is potentially unstable (i.e., the drone control tasks). In addition, PPO was the algorithm for which the modifications showed the most promise. We are thus currently running hyperparameter sweeps for the additive variant on the trajectory tracking and drone racing task, and will add comparisons to the absolute actions in the appendix once the runs are complete.
>
> ### Paper updates
> In response to your suggestions, we have made these changes to the paper:
> - Section 2 now includes a discussion on the zero-centeredness of delta actions and how to correct for them (2.4.1).
> - We ran all algorithms on the idealized environment for the modified delta quaternion and matrix networks, including both additive and scaled variants.
> - Section 3 has a newly added hypothesis (4) on the zero-centeredness of delta orientations, where we discuss the points you raised.
> - We find that the addition is helpful mainly for PPO, and can sometimes be detrimental (SAC).
> - We added the complete set of experiments to the appendix (A.3).
> - We added a discussion of the implementation variants to the appendix.
>
> ## Minor clarifications
>
> While minor, we gathered from your review that it was not entirely clear how the stochastic mappings are projected onto the manifold. We thus also clarified the last two paragraphs in Section 2.4.
>
> Finally, we want to emphasize that while scaling the local tangent vectors improves stability, we found it to be fairly robust even without scaling. Figure 10 and 11 in A.1.5 show the worst five runs out of 50. Even then, the unscaled runs converge 48 out of 50 times for SAC with sparse rewards, and always converge for TD3. So while you are correct that all other delta representations require special treatment, tangent spaces mostly work out of the box.
>
> ## Summary
> Please let us know if the modifications address your concerns, especially regarding the centering around the unit rotation. And thank you once again for the thoughtful review and suggestions!

---

> > ### Author Response · Authors · 2025-11-27
> > **Completion of additional results**
> >
> > Dear reviewer,
> >
> > we have completed our additional experiments and uploaded an updated version of the paper. As mentioned before, you can find the differences to the previous version marked in red. Adding to the results we mentioned in the previous comment, section A.3 now also includes results for the drone trajectory tracking and drone racing task.
> >
> > As suspected, the unit-centered versions of both quaternion and matrix representations improve significantly on these tasks. This is particularly true for the matrix representation. The improvement is more decisive for the tracking task, but still very much visible for the drone racing benchmark as well. As before, the unit-centered rotations do not quite achieve the performance of the relative tangent space representation, although the centered matrix representation comes close on the trajectory task.
> >
> > As a side note, we now also include a comparison against the 6D representation of Zhou et al. [1] to address the criticism of reviewer gmAW. If you are interested, you can find the results in A.4.
> >
> > Even though you have already indicated that you will raise your score, we would be much obliged if you could briefly let us know if you are still missing something from our revision of the paper. Otherwise, we once again thank you for the thoughtful review and our discussion.
> >
> > [1] Yi Zhou, Connelly Barnes, Lu Jingwan, Yang Jimei, and Li Hao. On the Continuity of Rotation Representations in Neural Networks. In The IEEE Conference on Computer Vision and Pattern Recognition (CVPR), June 2019.

---

> ### Comment · Reviewer_a6NH · 2025-11-21
> **Regarding primary concern: Quaternions / Rotation matrices not centered around the unit rotation**
>
> Your first suggestion of "leaving the network output range as is ($[-1, 1]$) and add the unit rotation to the output" sounds sensible. Considering that you want to model changes in rotation which typically do not deviate to much from the unit rotation, I am positive that not being able to model quaternions with negative scalar part is not a problem. The same applies for rotation matrices, given the identity rotation (3x3 unit matrix), the changes in rotation will also only marginally deviate from this rotation. Of course, this depends on the specific problem setting at hand. Alternatively, you could also multiply your output head by factor two to cover full SO(3).
>
> I appreciate the changes made. After checking your revised paper, I will increase my score.

---

> > ### Author Response · Authors · 2025-11-21
> >
> > Regarding the alternative output factor multiplication: The second formulation that we mentioned in our response does that, but only scales the network outputs that were biased by adding the unit element so that the off-diagonal (matrix) or xzy components (quaternions) cannot overshoot. One could also try to scale the full head for approximately equal sensitivities, but we don't expect a major change from the scaling version we presented.
> >
> > For the paper, we now compare both the additive and the scaled variant against the baseline. As previously mentioned, we are still running the hyperparameter sweeps and experiments for the drone benchmarks. These will be added in the appendix, and we will reflect the results in the discussion of Section 4. We expect this to take another week and will notify you when the last results are integrated in the paper if that is okay with you.
> >
> > Thank you once again for the review and the very specific suggestions on how to improve the paper!

---

> > ### Author Response · Authors · 2025-11-22
> > **Text highlighting**
> >
> > We have now highlighted changes we made in the paper in red to make them easily visible to reviewers.

---

### Official Review · Reviewer_1zXm · 2025-11-01

**Soundness:** 3
**Presentation:** 2
**Contribution:** 3
**Rating:** 6
**Confidence:** 4

**Summary:**

This paper studies how the representation of rotation in actions affects RL training. The authors compare different types of SO(3) action representation in a range of RL environments, including an ideal rotation control environment and some more realistic robotic benchmarks. From the results, the authors recommend using tangent vectors for practitioners.

**Strengths:**

1. The SO(3) action representation is a commonly encountered problem by RL practitioners, especially in the field of robotic control. This work provides valuable guidance for the design of the action space and environments.

2. The impact of different rotation representations is studied in a diverse set of benchmarks, and all the hypotheses are accompanied by detailed analysis. They all make the conclusions convincing.

**Weaknesses:**

1. The considered scope of this work is somewhat limited. Only the simple MLP network + Gaussian noise as policy representation is studied. Other popular design choices, such as discretizing actions into bins and learning a categorical distribution over the discrete action space could also be investigated.

2. The insights/takeaways from experiments could be better presented. In Section 3.3, the authors state several hypotheses but the conclusions are not highlighted. Since the results are quite mixed for different representations, rewards and algorithms, I think the presentation of the analysis could be improved.

**Questions:**

Please refer to the weaknesses part.

---

> ### Author Response · Authors · 2025-11-19
> **Response to Reviewer 1zXm**
>
> Dear reviewer,
>
> Thank you for the review. We are glad you find this paper a valuable resource for practitioners. In the following, we would like to address the two issues you raised.
>
> ## Presentation of the key findings
>
> We agree that the paper should highlight key takeaways, especially since one of its main points is to guide practitioners. We have used the additional space of the rebuttal to further emphasize our key takeaways from Section 3.3. The core learnings are now explicitly reiterated after every hypothesis, and use a background highlighter to make them stand out visually.
>
> We have also added a section on relative actions with quaternion and matrix representations (see answer to reviewer a6NH), connected the findings to the results from Section 4, and improved the Robosuite plot in Section 4 (see response to reviewer gmAW). We hope this resolves your concerns regarding the presentation of the analysis.
>
> ## Missing discussion on discretized action spaces
>
> Your second concern centers around the scope of the work. As correctly pointed out, the paper focuses on continuous control algorithms. While we agree that it would be interesting to investigate algorithms such as DQN with discrete action spaces, much of the analysis is based on representation continuities, exploration under projections, and entropy regularization. A thorough analysis of representations for discrete-action algorithms would need to cover entirely different topics, such as discretization schemes and the required density of the SO(3) cover, among others. These topics warrant dedicated sections and far exceed the scope of what can be covered in a single paper.
>
> We hope that you agree with us that a primer with a singular focus on continuous actions is valuable to the community, and that it makes sense to address the fundamentally different challenges of discrete-action algorithms in future work. We have added a corresponding remark on discretized action spaces to the limitations section of our paper.
>
> ## Summary
>
> We believe our changes to the presentation of results and highlighting of key findings address your concerns regarding the presentation, and have explicitly added an acknowledged of the limitation to continuous actions. Please let us know if you feel something was not appropriately addressed. Otherwise, we hope our clarifications improve your assessment of the paper.
>
> Edit: We have now highlighted changes we made in the paper in red to make them easily visible to reviewers.

---

> ### Author Response · Authors · 2025-11-27
>
> Dear reviewer,
>
> we have completed our additional experiments and uploaded an updated version of the paper. You can find the differences to the previous version marked in red. In summary, we have
> - improved the presentation significantly following your advice
> - improved the analysis and presentation of results in section 4
> - included discrete action spaces into our limitation section
>
> In addition, we added more experimental results following the remarks of reviewer a6NH and reviewer gmAW. These experiments are now finished and integrated into the paper.
>
> We would be very grateful if you could let us know if our improvements sufficiently addressed your criticism of our work, and if so if you would consider this in your score. Otherwise, please let us know where the updated version is still lacking.

---

### Author Response · Authors · 2025-12-01
**Summary of the Rebuttal Phase**

Dear AC and reviewers,

since reviewers are no longer able to respond to our rebuttals due to the recent OpenReview incident and ACs have been reassigned, we want to summarize the review and discussion until this point.

## Reviewer 1zXm
The review highlights that the paper discusses a commonly encountered problem and provides valuable guidance, with a diverse set of benchmarks and convincing conclusions.

### Criticism
The two main points of critic in this reviewer were that 1) the paper does not discuss discrete action spaces and 2) the presentation of the analysis could be improved.

### Response
We now summarize our main takeaways once more in a highlighted box beneath each hypothesis and have improved the clarity of our analysis. We also now mention discrete action spaces as a limitation and future work, since possible problems with discretized SO(3) action spaces are fundamentally different from the continuous case, which is our main focus.

### Decision
While we addressed the points raised on the presentation and argued that discrete actions are out of scope, the reviewer could not respond to confirm if their expectations were met and if they follow our reasoning regarding discrete action spaces due to the incident.

## Reviewer a6NH
The review remarks that the paper looks at a critical issue for the reinforcement learning community that so far has been mostly overlooked. It commends the extensive experiments, and judges the paper "an excellent fit for ICLR".

### Criticism
The main point of criticism focused on the relative matrix and quaternion action representation. The reviewer suspected that representations that are centered around the unit rotation would lead to a significant performance improvement. If we were able to cover this case in detail and provide experimental results on this additional variant, reviewer a6NH promised to **raise the rating to a strong accept (10) with absolute certainty (5)**.

### Response
We now cover the topic raised by the reviewer in an additional subsection in our theoretical chapter (2.4.1), added the hypothesis to chapter 3 (hypothesis 4), ran extensive experiments (including hyperparameter tuning) for PPO, SAC and TD3 on the idealized environment, and discuss the results in detail. Furthermore, and exceeding the requirements of the reviewer, we also ran experiments on the drone benchmarks. The full results and additional ablations and discussions on implementation details have been added in the appendix A.3.

### Decision
**The reviewer confirmed that we answered all questions to their satisfaction and promised to increase the score after reading the revised paper before the incident.**

## Reviewer gmAW
The review remarks that the paper studies an important topic which has not been covered yet. Prior studies on rotation representations have been conducted in the context of supervised learning, but not in RL. It states that the hypotheses point to interesting explanations of the advantages of some representations.

### Criticism
The primary issue identified in this review was the **missing comparison with the 6D representation introduced by Zhou et al.** [1]. It requested to add comparisons for the representation, and perhaps for its relative counterpart. In addition, the review posed several minor questions on presentation and asked for clarifications.

### Response
In response to the review, **we conducted all experiments on the idealized environment for all algorithms on the 6D representation**. We emphasized that we did initially think about including the 6D representation, but judged it similar to the rotation matrix representation based on Geist et al. [2]. Appendix A.4 now explicitly discusses this intuition, compares the performance for all algorithms and shows that the two representations indeed behave largely similar. Since the 6D representation only improves over the matrix representation for PPO, albeit marginally, we also run the PPO robot benchmarks and show that the impact on real tasks is inconclusive as expected.

We also addressed all questions and requests for clarifications by updating table 1, improving figure 2 and 3's descriptions, and adding a summary bar to figure 4.

### Decision
While we addressed all raised points, and the missing baseline in particular, the reviewer could not respond to confirm their expectations were met due to the incident.

[1] Y. Zhou, C. Barnes, J. Lu, J. Yang and H. Li, "On the Continuity of Rotation Representations in Neural Networks," 2019 IEEE/CVF Conference on Computer Vision and Pattern Recognition (CVPR).

[2] A. Ren´e Geist, Jonas Frey, Mikel Zhobro, Anna Levina, and Georg Martius. Learning with 3D rotations: a Hitchhiker’s guide to SO(3). In Proceedings of the 41st International Conference on Machine Learning, 2024

---

### Meta-Review · Area_Chair_vRMP · 2026-01-06

**Summary:**

1. *1zXm* limited scope in terms of models and action distribution parameterization
2. *1zXm* result presentation is unclear
3. *a6NH* each rotational representation requires special architectural and normalization considerations which are not adequately addressed and complicate the comparison
4. *gmAW* notes an important missing rotational representation

**Reviewer Concerns:**

1. the authors believe their scope is appropriate
2. the authors clarified the important conclusions
3. the authors performed additional experiments to evaluate the impact of these choices.
4. the authors added this representation to the paper.

**Reviewer Scores:**

- *1zXm* gave 6 and there is a small chance they may have increased their score.
- *a6NH* explicitly noted they would increase their score from 6 to 8.
- *gmAW* There is a good chance they may have increased their score from 4 to 6.

---

### Decision · Program_Chairs · 2026-01-26

Accept (Poster)